# Comparative interactome analysis of α-arrestin families in human and *Drosophila*

Kyung-Tae Lee[1,2†], Inez KA Pranoto[3†], Soon-Young Kim[4], Hee-Joo Choi[5,6,7], Ngoc Bao To[1], Hansong Chae[1], Jeong-Yeon Lee[5,6], Jung-Eun Kim[4], Young V Kwon[3]*, Jin-Wu Nam[1,2,5]*

[1]Department of Life Science, College of Natural Sciences, Hanyang University, Seoul, Republic of Korea; [2]Hanyang Institute of Advanced BioConvergence, Hanyang University, Seoul, Republic of Korea; [3]Department of Biochemistry, University of Washington, Seattle, United States; [4]Department of Molecular Medicine, Cell and Matrix Research Institute, School of Medicine, Kyungpook National University, Daegu, Republic of Korea; [5]Bio-BigData Center, Hanyang Institute for Bioscience and Biotechnology, Hanyang University, Seoul, Republic of Korea; [6]Department of Pathology, College of Medicine, Hanyang University, Seoul, Republic of Korea; [7]Hanyang Biomedical Research Institute, Hanyang University, Seoul, Republic of Korea

*For correspondence:
ykwon7@uw.edu (YVK);
jwnam@hanyang.ac.kr (JWN)

†These authors contributed equally to this work

Competing interest: The authors declare that no competing interests exist.

**Abstract** The α-arrestins form a large family of evolutionarily conserved modulators that control diverse signaling pathways, including both G-protein-coupled receptor (GPCR)-mediated and non-GPCR-mediated pathways, across eukaryotes. However, unlike β-arrestins, only a few α-arrestin targets and functions have been characterized. Here, using affinity purification and mass spectrometry, we constructed interactomes for 6 human and 12 *Drosophila* α-arrestins. The resulting high-confidence interactomes comprised 307 and 467 prey proteins in human and *Drosophila*, respectively. A comparative analysis of these interactomes predicted not only conserved binding partners, such as motor proteins, proteases, ubiquitin ligases, RNA splicing factors, and GTPase-activating proteins, but also those specific to mammals, such as histone modifiers and the subunits of V-type ATPase. Given the manifestation of the interaction between the human α-arrestin, TXNIP, and the histone-modifying enzymes, including HDAC2, we undertook a global analysis of transcription signals and chromatin structures that were affected by TXNIP knockdown. We found that TXNIP activated targets by blocking HDAC2 recruitment to targets, a result that was validated by chromatin immunoprecipitation assays. Additionally, the interactome for an uncharacterized human α-arrestin ARRDC5 uncovered multiple components in the V-type ATPase, which plays a key role in bone resorption by osteoclasts. Our study presents conserved and species-specific protein–protein interaction maps for α-arrestins, which provide a valuable resource for interrogating their cellular functions for both basic and clinical research.

## eLife assessment

This study provides a **valuable** resource that documents the protein–protein interactions (PPI) network for α-arrestins in both human and *Drosophila* based on affinity purification/mass spectrometry and the SAINTexpress method followed by a series of bioinformatic and functional assessments. Through these, the authors confirmed the roles of known and novel interactions, including proteins involved in RNA splicing and helicase, GTPase-activating proteins, and ATP synthase. This study

represents a **convincing** example of how to adopt comparative molecular interactions and how to interpret the functional implications.

## Introduction

The discovery of first arrestin protein in retinal rods contributed to a deeper understanding of photoreceptor signaling mediated by rhodopsin, which is one of the G-protein-coupled receptor (GPCR) class, and after its ability to arrest the GPCR signaling pathway, the protein was first named as 'arrestin' (*Kühn et al., 1984*; *Wilden et al., 1986*; *Zuckerman and Cheasty, 1986*). Shortly after this discovery of the first arrestin protein in the retina, another arrestin protein that specifically turns off β-adrenergic signaling, another type of GPCR, through 'receptor desensitization' was identified and named 'β-arrestin' (*Benovic et al., 1989*; *Lohse, 1992*; *Shenoy and Lefkowitz, 2011*). Further studies have revealed that β-arrestins regulate the receptor desensitization of other signaling pathways through ubiquitination and regulation of trafficking of various cargo molecules (*Kim and Benovic, 2002*; *Malik and Marchese, 2010*; *Puca and Brou, 2014*).

Another class of arrestin, α-arrestin, was first studied in fungi and yeast (*Andoh et al., 2002*) and subsequently recognized as a new class of arrestins (*Boase and Kelly, 2004*; *Herranz et al., 2005*). They contain characteristic arrestin domains, arrestin_N and arrestin_C, and PPxY motifs, which are unique to the α-arrestin clan (*Puca and Brou, 2014*). A phylogenetic study of arrestin proteins showed that α-arrestins are the ancestral class of the arrestin family and conserved from yeast to human (*Alvarez, 2008*). To date, six α-arrestins, arrestin domain containing protein 1 (ARRDC1), ARRDC2, ARRDC3, ARRDC4, ARRDC5, and thioredoxin-interacting protein (TXNIP), have been found to be in the human genome (*Zbieralski and Wawrzycka, 2022*). These human α-arrestins were first studied in conjunction with β-arrestins in the regulation of the β2-adrenergic receptor (β2AR) in human cells. ARRDC3 and ARRDC4 work as an adaptor protein for the ubiquitination of β2AR by recruiting the NEDD4 protein, an E3 ubiquitin ligase, through its conserved PPxY motifs (*Han et al., 2013*; *Nabhan et al., 2010*; *Shea et al., 2012*). In addition to their β2AR-associated roles, α-arrestins are involved in trafficking and sorting of other GPCRs and signaling molecules through post-translational modifications, including ubiquitination. For example, ARRDC1 and ARRDC3 were reported to play roles in the degradation of the Notch receptor (*Puca et al., 2013*) and in the ubiquitination of ALG-2-interacting protein X (ALIX) (*Dores et al., 2015*). Uniquely, ARRDC1 have been reported to mediate microvesicle budding by recruiting E3 ligases, such as WW domain-containing E3 ubiquitin protein ligase2 (WWP2). This recruitment leads to its own ubiquitination. Additionally, ARRDC1 possesses a PSPA motif that binds the tumor susceptibility gene 101 (TSG101) protein, an essential component of an endosomal sorting complex that is also required for this ARRDC1-mediated microvesicle budding (*Nabhan et al., 2012*). Another well-known α-arrestin, TXNIP, was first named as vitamin D3-upregulated protein 1 (VDUP1) after verification that its gene is a vitamin D3 target in cancer cells (*Chen and DeLuca, 1994*; *Qayyum et al., 2021*). Since then, TXNIP had been reported to directly interact with thioredoxin, which is an essential component of the cellular redox system, to inhibit its activity as an antioxidant (*Junn et al., 2000*; *Nishiyama et al., 1999*; *Patwari et al., 2006*). TXNIP was also reported to inhibit glucose uptake by inducing the internalization of glucose transporter 1 (GLUT1) through clathrin-mediated endocytosis and by indirectly reducing GLUT1 RNA levels (*Wu et al., 2013*). Although the TXNIP is known to be localized in both cytoplasm and nucleus, biological functions of TXNIP have been mostly explored in cytoplasm but remained poorly characterized in nucleus.

A few α-arrestins appear to have evolutionarily conserved functions in both human and invertebrates. For instance, the Hippo signaling pathway, which impacts a variety of cellular processes such as metabolism, development, and tumor progression (*Mo et al., 2014*; *Pei et al., 2015*; *Schütte et al., 2014*; *Wang et al., 2010*; *Zhi et al., 2012*), was shown to be regulated by α-arrestin in both *Drosophila* (*Kwon et al., 2013*) and human cells (*Xiao et al., 2018*). In *Drosophila*, the protein Leash was identified as an α-arrestin and shown to downregulate Yki by promoting its lysosomal degradation, leading to a restriction in growth (*Kwon et al., 2013*). In human cells, ARRDC1 and ARRDC3 were shown to induce degradation of the mammalian homolog of Yki, YAP1, by recruiting the E3 ubiquitin ligase ITCH in renal cell carcinoma (*Xiao et al., 2018*), suggesting functional homology between human and *Drosophila*. However, because the α-arrestins interact with multiple targets, an unbiased, comparative analysis of interactome is required to determine whether other α-arrestin from human

and *Drosophila* have common and specific interacting partners, which will determine their functional homology and diversification.

A comprehensive understanding of their protein–protein interactions (PPIs) and interactomes will shed light on the underlying molecular mechanisms, reveal novel regulatory axes, and enable the identification of previously unrecognized roles of α-arrestin in cellular processes. Furthermore, extensive characterization of the α-arrestin interactomes may help uncover potential therapeutic targets and provide valuable insights into the treatment of diseases associated with dysregulated signaling pathways (*Diaz et al., 2005*; *Lu et al., 2008*; *Wang et al., 2018*; *Zhou et al., 2010*). However, to date, no comprehensive and comparative analysis of PPIs associated with α-arrestins has been conducted.

In this study, we conducted affinity purification/mass spectrometry (AP/MS) of 6 human and 12 *Drosophila* α-arrestins. A high-confidence PPI network was constructed by selecting a cutoff for receiver operating characteristic (ROC) curves of Significance Analysis of INTeractome express (SAINTexpress) scores (*Teo et al., 2014*). The constructed interactomes were validated using known affinities between domains of prey proteins and the short-linear motifs of α-arrestins. We also investigated orthologous relationships between binding partners from human and *Drosophila* and found that many proteins with both known and novel functions could be conserved between two species. Finally, we performed experiments to provide new insights into the functions of TXNIP and ARRDC5 that were revealed in our study. Together, our results provide a valuable resource that describes the PPI network for α-arrestins in both human and *Drosophila* and suggest novel regulatory axes of α-arrestins.

## Results

### High-confidence α-arrestin interactomes in human and *Drosophila*

Genome-scale sets of prey proteins interacting with α-arrestins (referred to herein as 'interactomes') were compiled by conducting AP/MS for 6 human and 12 *Drosophila* α-arrestin proteins (*Figure 1A*, *Supplementary file 1*). Proteins possibly interacting with α-arrestins were pulled down from total cell lysates of human embryonic kidney 293 (HEK293) and S2R+ cells stably expressing GFP-tagged α-arrestins (*Figure 1B*, *Figure 1—figure supplement 1*). All α-arrestin experiments were replicated twice, and negative control experiments were conducted multiple times. In total, 3243 and 2889 prey proteins involved in 9908 and 13,073 PPIs with human and *Drosophila* α-arrestins, respectively, were initially detected through AP/MS (*Figure 1—source data 1*).

To build high-confidence interactomes of α-arrestin family proteins, a probabilistic score for individual PPIs was estimated using SAINTexpress (*Teo et al., 2014*) and an optimal cutoff for the scores was set using positive and negative PPIs of α-arrestin from public databases and the literature (*Colland et al., 2004*; *Dotimas et al., 2016*; *Draheim et al., 2010*; *Mellacheruvu et al., 2013*; *Nabhan et al., 2012*; *Nishinaka et al., 2004*; *Puca and Brou, 2014*; *Szklarczyk et al., 2015*; *Warde-Farley et al., 2010*; *Wu et al., 2013*; *Supplementary file 2*). The resulting ROC curves showed high area under the curve (AUC) values and the SAINTexpress scores at which the false discovery rate (FDR) was 0.01 were selected as cutoffs (0.85 for human and 0.88 for *Drosophila*, *Figure 1C*). Given the cutoffs, 1306 and 1732 PPIs involving 902 and 1732 proteins were selected for human and *Drosophila*, respectively. Because proteins of low abundance (low spectral counts) are easily affected by a stochastic process (*Lundgren et al., 2010*; *Old et al., 2005*), the minimum spectral count of PPIs was set at 6, allowing us to select PPIs with higher confidence. In fact, the spectral counts of the filtered PPIs were highly reproducible between replicates (*Figure 1—figure supplement 2A*; Pearson's correlations, 0.91 for human; 0.89 for *Drosophila*) and principal component analysis (PCA) based on $\log_2$ spectral counts also confirmed a high reproducibility between replicates (*Figure 1—figure supplement 2B*). As a result, we successfully identified many known interaction partners of α-arrestins such as NEDD4, WWP2, WWP1, ITCH, and TSG101, previously documented in both literatures and PPI databases (*Figure 1—figure supplement 2C–F*; *Colland et al., 2004*; *Dotimas et al., 2016*; *Draheim et al., 2010*; *Mellacheruvu et al., 2013*; *Nabhan et al., 2012*; *Nishinaka et al., 2004*; *Puca and Brou, 2014*; *Szklarczyk et al., 2015*; *Warde-Farley et al., 2010*; *Wu et al., 2013*). Additionally, we greatly expanded repertoire of PPIs associated with α-arrestins in human and *Drosophila*, resulting in 390 PPIs between 6 α-arrestins and 307 prey proteins in human, and 740 PPIs between 12 α-arrestins and 467 prey proteins in *Drosophila* (*Figure 1—figure supplement 2E*). These are subsequently referred to as 'high-confidence PPIs' (*Supplementary file 3*).

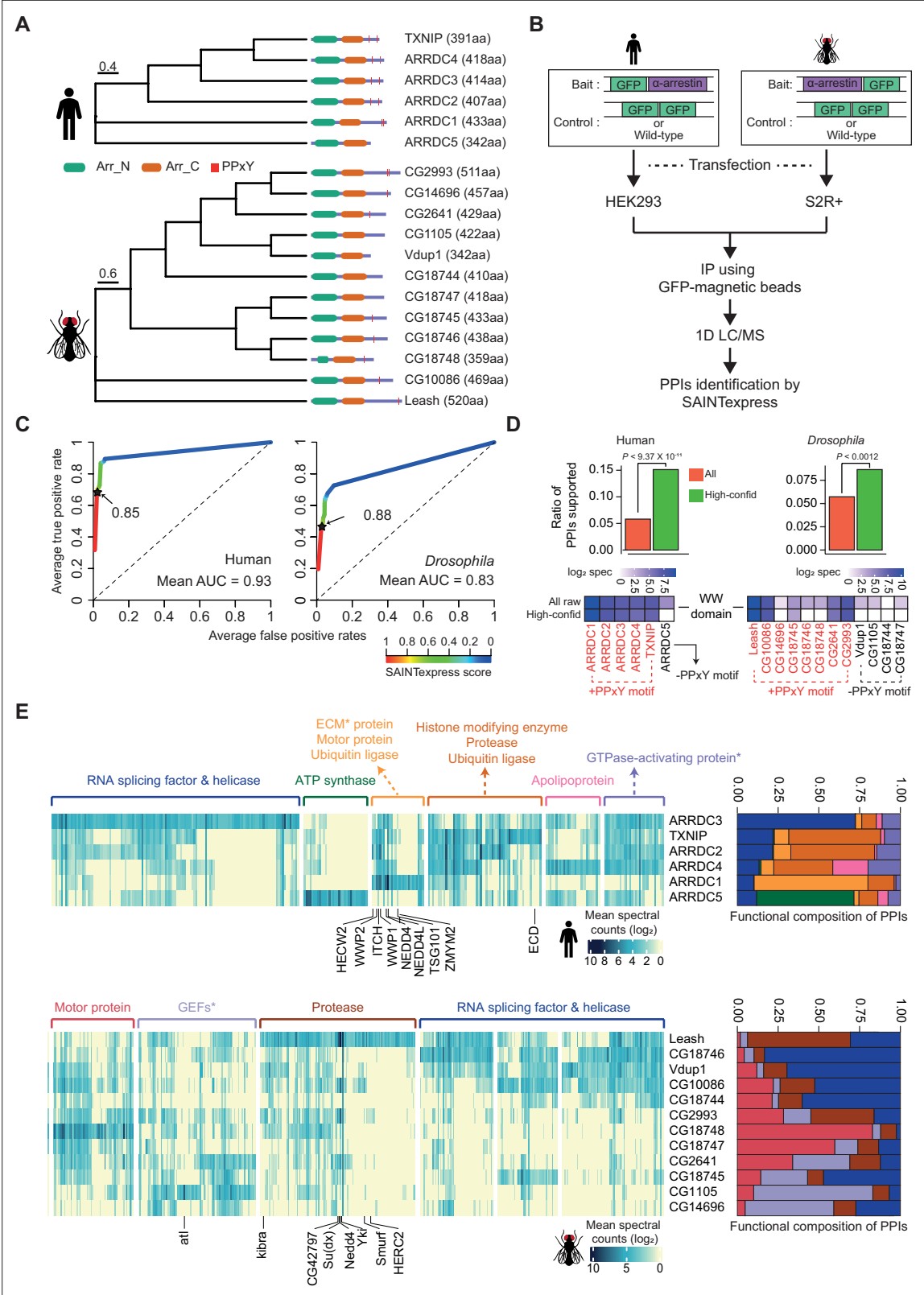

**Figure 1.** Identification of high-confidence α-arrestin protein–protein interactions (PPIs). (**A**) Phylogenetic tree of α-arrestins from human (6, top) and *Drosophila* (12, bottom) based on protein sequences. The numbers in parentheses indicate the length of each protein. aa, amino acids; Arr_N: Arrestin N domain; Arr_C: Arrestin C domain; PPxY: PPxY motif. (**B**) Shown is a schematic flow of affinity purification/mass spectrometry (AP/MS) experiments and computational analysis. (**C**) Receiver operating characteristic (ROC) curves of Significance Analysis of INTeractome express (SAINTexpress) scores along

*Figure 1 continued*

with mean area under the curve (AUC) values. The arrows point to the cutoff scores used in subsequent studies in human (left) and *Drosophila* (right). (**D**) Top: the fraction of 'high-confid' (high-confidence) and raw (unfiltered) PPIs that are supported by known affinities between short-linear motifs and protein domains in human (left) and *Drosophila* (right). One-sided, Fisher's exact test was performed to test the significance. Bottom: the sum of $\log_2$ spectral counts ('$\log_2$ spec') of proteins with WW domains that were reported to interact with each α-arrestin in the high-confidence or raw PPI sets are depicted as heatmap. (**E**) The α-arrestins and their interactomes were hierarchically clustered based on the $\log_2$ mean spectral counts and summarized for human (top) and *Drosophila* (bottom) in the heatmaps. The functionally enriched protein class in the clustered interactomes are indicated on the top. Proteins that were reported to interact with α-arrestins in literatures and databases are selectively labeled on the bottom. On the right, the functional composition of the clustered α-arrestin interactomes is summarized as the sum of $\log_2$ mean spectral counts, which are colored to correspond with the labels on the left.

The online version of this article includes the following source data and figure supplement(s) for figure 1:

**Source data 1.** Raw spectral count tables.

**Figure supplement 1.** Fluorescence images showing HEK293 and S2R+ cells stably expressing GFP-tagged α-arrestins.

**Figure supplement 2.** Affinity purification/mass spectrometry (AP/MS) data offer high reproducibility and expand the protein–protein interactions (PPIs) associated with α-arrestins while also reaffirming known interactions.

**Figure supplement 3.** Protein domains and subcellular localization of α-arrestin interactomes.

## Short-linear motifs and protein domains enriched in α-arrestins and their interactomes

To validate our high-confidence PPIs, we sought to analyze known short-linear motifs in α-arrestins, which are commonly 3–15 stretches of amino acids that are known to participate in interactions with other protein domains (*Dinkel et al., 2016*). Utilizing the known affinities between short-linear motifs in α-arrestins and protein domains in interactomes (*El-Gebali et al., 2019*; *The UniProt Consortium, 2017*) from eukaryotic linear motif (ELM) database (*Dinkel et al., 2016*), we evaluated whether our high-confidence PPIs could be explained by the known affinities between them (*Supplementary file 4*). The fractions of our high-confidence PPIs (green, *Figure 1D*, top), supported by the known affinities, were significantly greater than those of all raw PPIs (red, *Figure 1D*, top) in both species ($p < 9.37 \times 10^{-11}$ for human and $p < 0.0012$ for *Drosophila*, one-sided Fisher's exact test, *Figure 1D*, top). One of the most well-known short-linear motifs in α-arrestin is PPxY, which is reported to bind with high affinity to the WW domain found in various proteins, including ubiquitin ligases (*Ingham et al., 2004*; *Macias et al., 1996*; *Sudol et al., 1995*). Our analysis revealed the specific enrichment of WW domain-containing proteins in the interactomes of α-arrestins with at least one PPxY motif but not in that of the human α-arrestin (ARRDC5) without a PPxY motif (*Figure 1D*, bottom-left). The interactomes of five out of the eight *Drosophila* α-arrestins with a PPxY motif were enriched for WW domain-containing proteins, but there was no such enrichment for any of the *Drosophila* α-arrestins without a PPxY motif (*Figure 1D*, bottom-right). In conclusion, a considerable portion of the high-confidence PPIs identified in this study can be evident by known affinities between short-linear motifs and protein domains.

Next, we conducted enrichment analyses of Pfam proteins domains (*El-Gebali et al., 2019*; *Huang et al., 2009b*) among interactome of each α-arrestin to investigate known and novel protein domains commonly or specifically associated (*Figure 1—figure supplement 3A*, *Supplementary file 5*). The most prominent interacting domains in both species were the Homologous to E6AP C-terminus (HECT), WW, and C2 domains (*Figure 1—figure supplement 3A*, *Supplementary file 5*). HECT and C2 domains are well known to be embedded in the E3 ubiquitin ligases such as NEDD4, HECW2, and ITCH along with WW domains (*Ingham et al., 2004*; *Melino et al., 2008*; *Rotin and Kumar, 2009*; *Scheffner et al., 1995*; *Weber et al., 2019*), and as we observed strong preference of WW domains to PPxY containing proteins (*Figure 1D*), these domains were significantly enriched in binding proteins of α-arrestins with PPxY motif in human and *Drosophila* (FDR < 0.033 – $1.23 \times 10^{-11}$ for human; FDR < 0.045 – $4.10 \times 10^{-6}$ for *Drosophila*, *Figure 1—figure supplement 3A*, *Supplementary file 5*). Other common protein domains involved in the protein degradation process, such as proteasome domains, were also significantly enriched in the interactomes (of ARRDC4 in human and Leash in *Drosophila*) in both species (FDR < $6.41 \times 10^{-4}$ for human and FDR < $1.30 \times 10^{-5}$ for *Drosophila*, *Figure 1—figure supplement 3A*, *Supplementary file 5*). Interestingly, some α-arrestins (ARRDC3 in human and Vdup1, Leash, and CG18746 in *Drosophila*) appeared to interact in common with RNA-binding domains,

such as DEAD box, helicase, WD40, and RNA recognition motif, but others did not. In addition, the cargo and motor protein domains IBN_N (FDR < 0.0076 for human and FDR < $2.50 \times 10^{-4}$ – $2.11 \times 10^{-6}$ for *Drosophila*) and myosin_head (FDR < 0.033 for human and FDR < $2.11 \times 10^{-6}$ for *Drosophila*) also interacted with several α-arrestins in common (ARRDC4 in human and CG1105, CG18745, and CG18748 in *Drosophila*, *Figure 1—figure supplement 3A*, *Supplementary file 5*). These enriched domains explain the conserved interactomes associated with RNA splicing and protein transport in both species. In addition, human α-arrestins seem to interact with human-specific domains, such as PDZ, Rho-GEF, MCM, laminin, zinc finger, and BAG6 domains, providing an expanded interactomes of human α-arrestins (*Figure 1—figure supplement 3A*, domains in black), indicating the presence of both conserved and specific protein domains interacting with α-arrestins.

## Expanded functional signatures of α-arrestin interactomes

Because the functions of α-arrestins can be inferred based on their binding partners, the prey proteins were grouped based on their interactions with α-arrestins, which revealed specialized functions of the respective α-arrestins with some redundancy as well as both known and novel functions (*Figure 1E*). The analysis of protein class enrichment by the PANTHER classification system (*Thomas et al., 2003*) revealed previously reported functions, such as 'Ubiquitin ligase' (FDR < 0.0019 and $5.01 \times 10^{-7}$ for human; Benjamini–Hochberg correction) and 'Protease' (FDR < $1.93 \times 10^{-6}$ for human and $5.02 \times 10^{-6}$ for *Drosophila*) (*Dores et al., 2015*; *Kwon et al., 2013*; *Nabhan et al., 2012*; *Puca et al., 2013*; *Rauch and Martin-Serrano, 2011*; *Shea et al., 2012*; *Xiao et al., 2018*). In fact, the known binding partners, NEDD4, WWP2, WWP1, and ITCH in human and CG42797, Su(dx), Nedd4, Yki, Smurf, and HERC2 in *Drosophila*, that were detected in our data are related to ubiquitin ligases and protein degradation (*Chen and Matesic, 2007*; *Ingham et al., 2004*; *Kwon et al., 2013*; *Marín, 2010*; *Melino et al., 2008*; *Rotin and Kumar, 2009*; *Figure 1E*, *Figure 1—figure supplement 2F*). In addition, novel biological functions of α-arrestins were uncovered. For instance, in human, prey proteins interacting with ARRDC3 displayed enrichment of 'RNA splicing factor and helicase' functions as well as 'GTPase-activating proteins', those of ARRDC4 were enriched with 'Apolipoprotein', and those of ARRDC5 with 'ATP synthase' (*Figure 1E*, up). Motor protein, protease, ubiquitin ligase, RNA splicing factor, and helicase were functions that were also enriched in *Drosophila* prey proteins (*Figure 1E*, bottom). Among them, the motor protein and RNA splicing, and helicase functions seemed to be novel conserved functions between human and *Drosophila*. The functional compositions of the interacting proteins summarized the common or highly specialized functions of α-arrestins well (*Figure 1E*, right panel). For example, in human, proteins that interacted with TXNIP, ARRDC2, and ARRDC4 showed similar ubiquitination and protease-related functions, whereas ARRDC3 and ARRDC5 displayed unique interactomes associated with other functions. For *Drosophila*, the interactomes of the [Vdup1, CG10086 and CG18744], [CG18748 and CG18747], and [CG1105 and CG14696] α-arrestin subsets each exhibited similar functional compositions, but the Leash interactome showed a distinct enrichment of ubiquitination-related and protease functions, consistent with prior reports highlighting Leash's role in the lysosomal degradation of hippo signaling pathway component, Yki (*Kwon et al., 2013*). Taken together, these results suggest that the resulting high-confidence PPIs of α-arrestins expanded the functional interactome maps of α-arrestins in both human and *Drosophila*.

## Subcellular localizations of α-arrestin interactomes

Cellular localizations of proteins often provide valuable information of their functions and activity, but only a small number of α-arrestins are known for their preferential subcellular localization. We thus examined the subcellular localizations of the interacting proteins using the cellular component feature in Gene Ontology (GO) using DAVID (*Huang et al., 2009a*; *Huang et al., 2009b*; *Figure 1—figure supplement 3B*, *Supplementary file 6*). Prey proteins (246 for human and 245 for *Drosophila*) that were localized in at least one cellular compartment were examined. We found that prey proteins of ARRDC5 were preferentially localized in the endoplasmic reticulum and at the plasma membrane but were less often localized in the nucleus compared to those of other human α-arrestins (*Figure 1—figure supplement 3B*, top). Similarly, the prey proteins of ARRDC1 and 4 were less often localized in the nucleus, instead being preferentially localized in the cytoplasm (ARRDC4) or extracellular space (ARRDC1), in agreement with previous reports (*Nabhan et al., 2012*; *Wang et al., 2018*). TXNIP seemed to preferentially interact with prey proteins in cytoplasm and nucleus (*Figure 1—figure*

*supplement 3B*, bottom), consistent with a previous report (*Kim et al., 2019a*; *Saxena et al., 2010*). ARRDC3, which was suggested to be localized in cytoplasm in previous study (*Nabhan et al., 2010*), appeared to interact with proteins preferentially localized in nucleus in addition to the ones in cytoplasm, implying novel functions of ARRDC3 in the nucleus. In *Drosophila*, the localization of interacting proteins is often uncharacterized compared to human, but a preference for a localization for part of the interactomes can be observed (*Figure 1—figure supplement 3B*, bottom). Some of them are preferentially localized at the plasma membrane (CG18747), mitochondria (CG14696), peroxisome (CG14696), lysosome (CG2641), or cytoskeleton (CG18748) compared to others. However, interactomes of Leash, Vdup1, CG2641, CG18745, CG18746, and CG10086 are preferentially localized in the nucleus. Taken together, these data about the preferential localizations of interacting proteins provide evidence about the functions and activity of α-arrestins in cells.

## Functional complexes in α-arrestin interactomes

The fact that protein functions are often realized in complexes (*Hartwell et al., 1999*) urged us to search for functional complexes that extensively interact with α-arrestins. For this analysis, protein complexes that are significantly connected with each α-arrestin were examined using the COMPlex Enrichment Analysis Tool (COMPLEAT) (*Vinayagam et al., 2013*), resulting in the detection of 99 and 18 protein complexes for human and *Drosophila*, respectively (*Supplementary file 7*). The complexes were iteratively combined with cellular components from GO (*Huang et al., 2009b*; *Supplementary file 7*) based on the overlap coefficients (*Vijaymeena and Kavitha, 2016*). The significance of the resulting combined complexes was then tested with the connectivity to each α-arrestin using the interquartile means (IQMs) of SAINTexpress scores compared to those from 1000 random cohorts. This approach showed that 33 clustered complexes comprising 335 protein subunits were significantly interacting with six human α-arrestins (*Figure 2*, *Figure 2—figure supplement 1*, FDR < 0.2) and 20 clustered complexes comprising 220 subunits were significantly interacting with *Drosophila* α-arrestins (*Figure 3*, *Figure 3—figure supplement 1*, FDR < 0.2).

The two largest complexes found to interact with α-arrestins were related to protein degradation (proteasome and ubiquitin-dependent proteolysis) and RNA splicing and processing in both species (*Figures 2 and 3*, *Figure 2—figure supplement 1*, *Figure 3—figure supplement 1*). ARRDC1, 2, and 4 and TXNIP in human and Leash and CG2993 in *Drosophila* were found to interact with protein degradation complexes. While the association of ARRDC3 with these ubiquitination-dependent proteolysis complexes is statistically insignificant, ARRDC3 does interact with individual components of these complexes such as NEDD4, NEDD4L, WWP1, and ITCH (*Figure 2—figure supplement 1*). This suggests their functional relevance in this context, as previously reported in both literature and databases (*Nabhan et al., 2010*; *Shea et al., 2012*; *Szklarczyk et al., 2015*; *Warde-Farley et al., 2010*; *Puca and Brou, 2014*; *Xiao et al., 2018*). On the other hand, ARRDC2 and 3 in human and Leash, CG18746, Vdup1, CG10086, and CG18744 in *Drosophila* were found to interact with RNA splicing and processing complexes. Although the above-mentioned α-arrestins interacted in common with the two complexes described above, they were also found to bind to distinct complexes. For instance, TXNIP specifically binds to transcriptional and histone deacetylase (HDAC) complexes, ARRDC1 to endosomal sorting and laminin complexes, ARRDC2 to the Set1C/COMPASS complex, ARRDC3 to transcription elongation factors (TEFs) and spindle assembly checkpoint (SAC) and cell polarity complexes, and ARRDC4 to clathrin-coated pit and BAT3 complexes in human. In *Drosophila*, Leash specifically binds to AP-2 adaptor and WASH complexes and CG18746 to the UTP B complex. ARRDC5 is specifically associated with V-type ATPase and vacuolar protein sorting complexes in human. CG18748 is associated with motor protein complexes, including actin, myosin, and microtubule-associated complexes in *Drosophila*. Taken together, the results from this analysis provide a glimpse of underexplored roles for α-arrestins in diverse cellular processes.

## Conserved interactomes of α-arrestins

Given that α-arrestins are widely conserved in metazoans (*Alvarez, 2008*; *DeWire et al., 2007*), we sought to exploit the evolutionarily conserved interactomes of human and *Drosophila* α-arrestins. For this analysis, we searched for orthologous relationships in the α-arrestin interactomes using the DRSC Integrative Ortholog Prediction Tool (DIOPT) (*Hu et al., 2011*). Among high-confidence prey proteins, 68 in human and 64 in *Drosophila* were reciprocally predicted to have ortholog relationships, defining

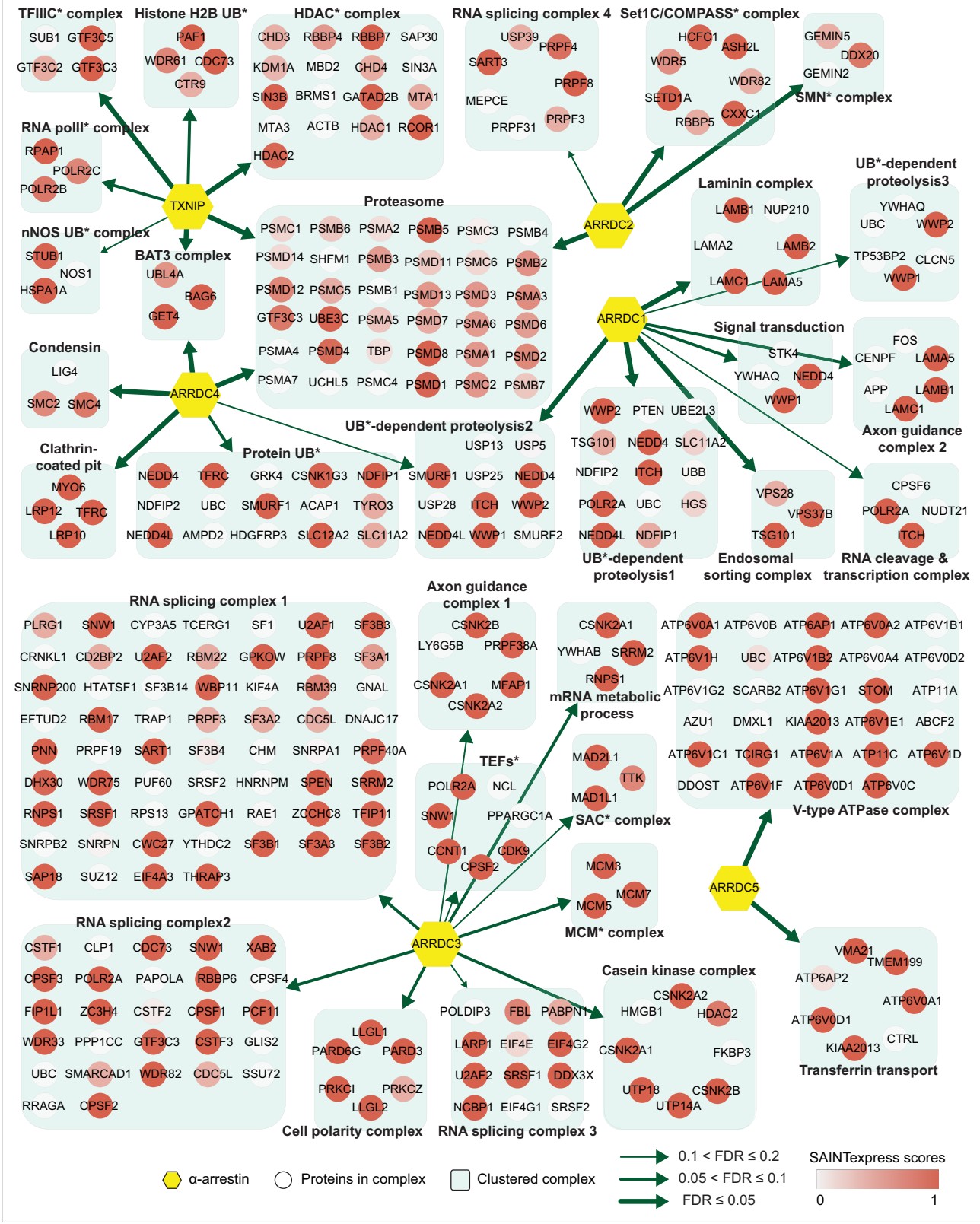

**Figure 2.** Network of α-arrestins and their associated protein complexes in human. Network of α-arrestins and the functional protein complexes that significantly interact with them in human. α-Arrestins are colored yellow and prey proteins in protein complexes are colored according to the Significance Analysis of INTeractome express (SAINTexpress) scores (averaged if the protein interacts with multiple α-arrestins). The thickness of the green arrows indicates the strength of the interaction between α-arrestins and the indicated protein complexes, wdhich was estimated with false

*Figure 2 continued on next page*

*Figure 2 continued*

discovery rate (FDR) of complex association scores (see 'Materials and methods'). UB, ubiquitination; HDAC, histone deacetylase; COMPASS, complex proteins associated with Set1; SMN, survivor of motor neurons; TFIIIC, transcription factor III C; RNA polII, RNA polymerase II; MCM, minichromosome maintenance protein complex.

The online version of this article includes the following figure supplement(s) for figure 2:

**Figure supplement 1.** Network of high-confidence protein–protein interactions (PPIs) between α-arrestins and individual proteins of its associated protein complexes in human.

58 orthologous prey groups (DIOPT score ≥ 2; *Supplementary file 8*). α-Arrestins were then hierarchically clustered based on the $\log_2$-transformed mean spectral counts of these orthologous interactome, defining seven groups of α-arrestins. Orthologous prey proteins were grouped according to their shared biological function, defining nine functional groups and others of diverse functions (*Figure 4*). The resulting clusters revealed PPIs that were functionally conserved. For instance, ARRDC3 in human and CG18746 in *Drosophila* actively interact with proteins in RNA binding and splicing groups. Leash in *Drosophila* appeared to interact with proteins in similar functional groups as ARRDC3 but, like ARRDC1, it also extensively interacts with members of ubiquitin-dependent proteolysis groups. In addition, ARRDC4 interacts with proteins in the motor protein and trafficking group, similar to CG18748 in *Drosophila*, and binds to proteins in the ubiquitin-dependent proteolysis group, similar to TXNIP. Similarly, CG10086 and Vdup1, CG14696 and ARRDC5, and CG2993 and ARRDC2 appeared to have conserved interactomes between human and *Drosophila*.

The most prominent functional modules shared across both species were the ubiquitin-dependent proteolysis, endosomal trafficking, and small GTPase-binding modules, which are in agreement with the well-described functions of α-arrestins in membrane receptor degradation through ubiquitination and vesicle trafficking (*Dores et al., 2015*; *Han et al., 2013*; *Kwon et al., 2013*; *Nabhan et al., 2012*; *Puca and Brou, 2014*; *Puca et al., 2013*; *Shea et al., 2012*; *Xiao et al., 2018*; *Zbieralski and Wawrzycka, 2022*; *Figure 4*). In contrast, the functional modules involving cyclin and cyclin-dependent kinase, casein kinase complex, and laminin seemed to be conserved between relatively specific sets of α-arrestins, whereas those related to motor proteins and RNA binding and splicing were more generally conserved. Taken together, the comparative analyses led us to identification of detailed, orthologous interactome maps of α-arrestins, which extend beyond the limited insights provided by sequence-based comparative analysis alone (*Figure 4—figure supplement 1*). Conserved roles of α-arrestins in both established and previously uncharacterized signaling pathways expand our understanding of the diverse roles of α-arrestins in cellular signaling.

## Chromatin accessibility is globally decreased under TXNIP depletion

TXNIP is one of the most well-studied α-arrestins. Previous studies reported that TXNIP interacts with transcriptional repressors, such as FAZF, PLZF, and HDAC1 or HDAC3, to exert antitumor activity (*Han et al., 2003*) or repress NF-kB activation (*Kwon et al., 2010*). However, although such studies provided information about interactions with a few transcriptional repressors, they barely provided a systematic view of the roles of TXNIP in controlling the chromatin landscape and gene expression. In that sense, our PPI analysis first revealed that TXNIP extensively binds to chromatin remodeling complexes, such as the HDAC and histone H2B ubiquitination complexes, as well as to transcriptional complexes, such as the RNA polymerase II and transcription factor IIIC complexes (*Figure 2*, *Figure 2—figure supplement 1*). Such PPIs indicate that TXNIP could control transcriptional and epigenetic regulators. To examine how the global epigenetic landscape is remodeled by TXNIP, we knocked down its expression in HeLa cells with a small interfering RNA (siTXNIP) and confirmed a decrease at both the RNA and protein levels (*Figure 5A and B*). We then produced two biological replicates of ATAC- and RNA-seq experiments in HeLa cells with TXNIP depletion (*Supplementary file 9*) to detect differentially accessible chromatin regions (dACRs) and differentially expressed genes (DEGs) (*Figure 5C*, *Supplementary file 10*). The replicated samples of both ATAC- and RNA-seq were well grouped in principal component spaces (*Figure 5—figure supplement 1A and B*). The normalized ATAC-seq signal and the RNA level of expressed genes clearly showed the enrichment of open chromatin signals around the transcription start sites (TSSs) of genes that are actively transcribed (*Figure 5—figure supplement 1C*). We detected 70,746 high-confidence accessible chromatin regions (ACRs) across all samples, most of which were located in gene bodies (38.74%), followed by intergenic regions (32.03%)

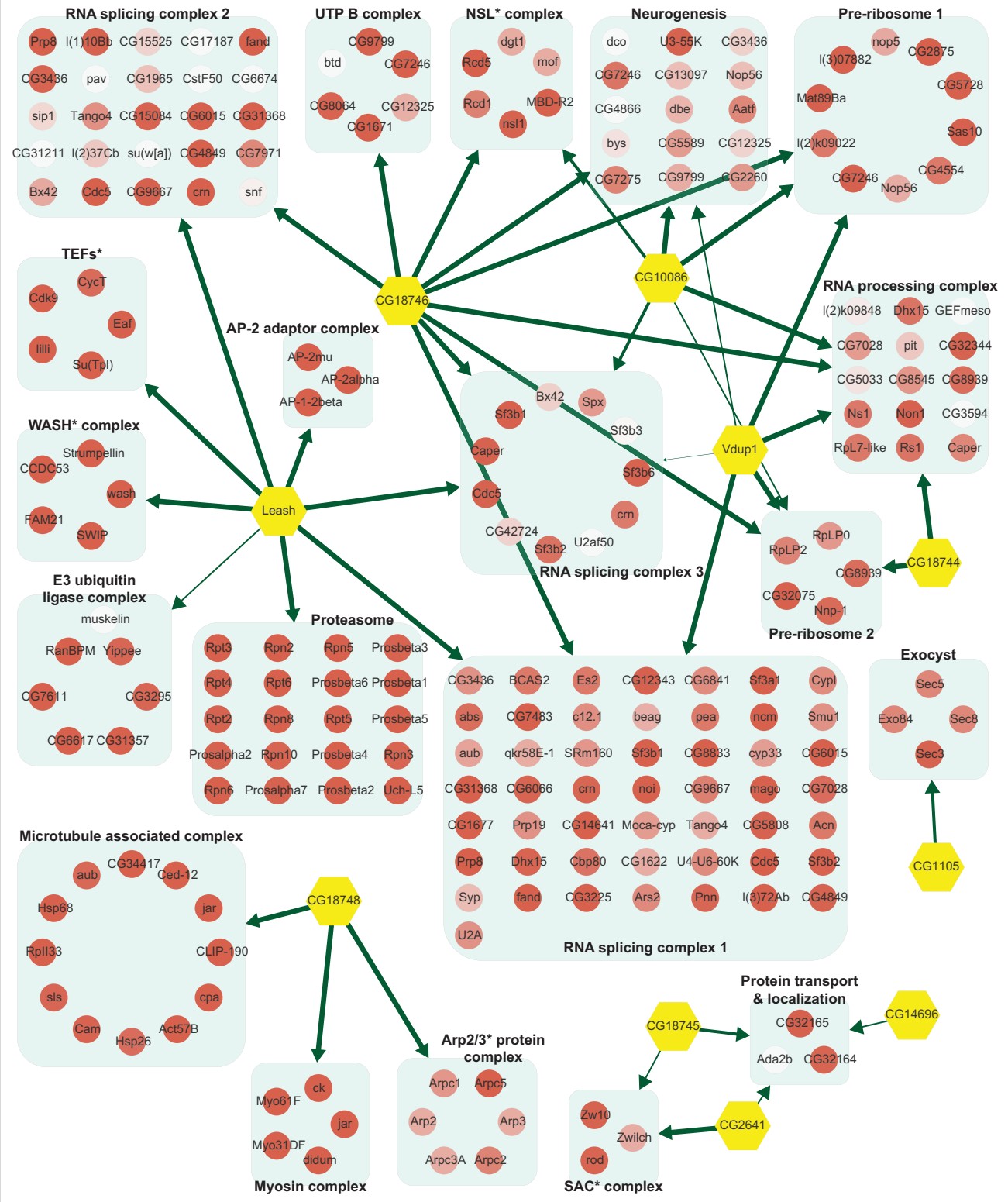

**Figure 3.** Network of α-arrestins and their associated protein complexes in *Drosophila*. Network of α-arrestins and the functional protein complexes that significantly interact with them in *Drosophila*, depicted in a manner analogous to *Figure 2*. SAC, spindle assembly checkpoint; NSL, non-specific lethal; WASH, Wiskott–Aldrich syndrome protein and scar homolog; Arp2/3, actin-related protein 2/3; TEF, transcription elongation factor.

The online version of this article includes the following figure supplement(s) for figure 3:

**Figure supplement 1.** Network of high-confidence protein–protein interactions (PPIs) between α-arrestins and individual proteins of its associated protein complexes in *Drosophila*.

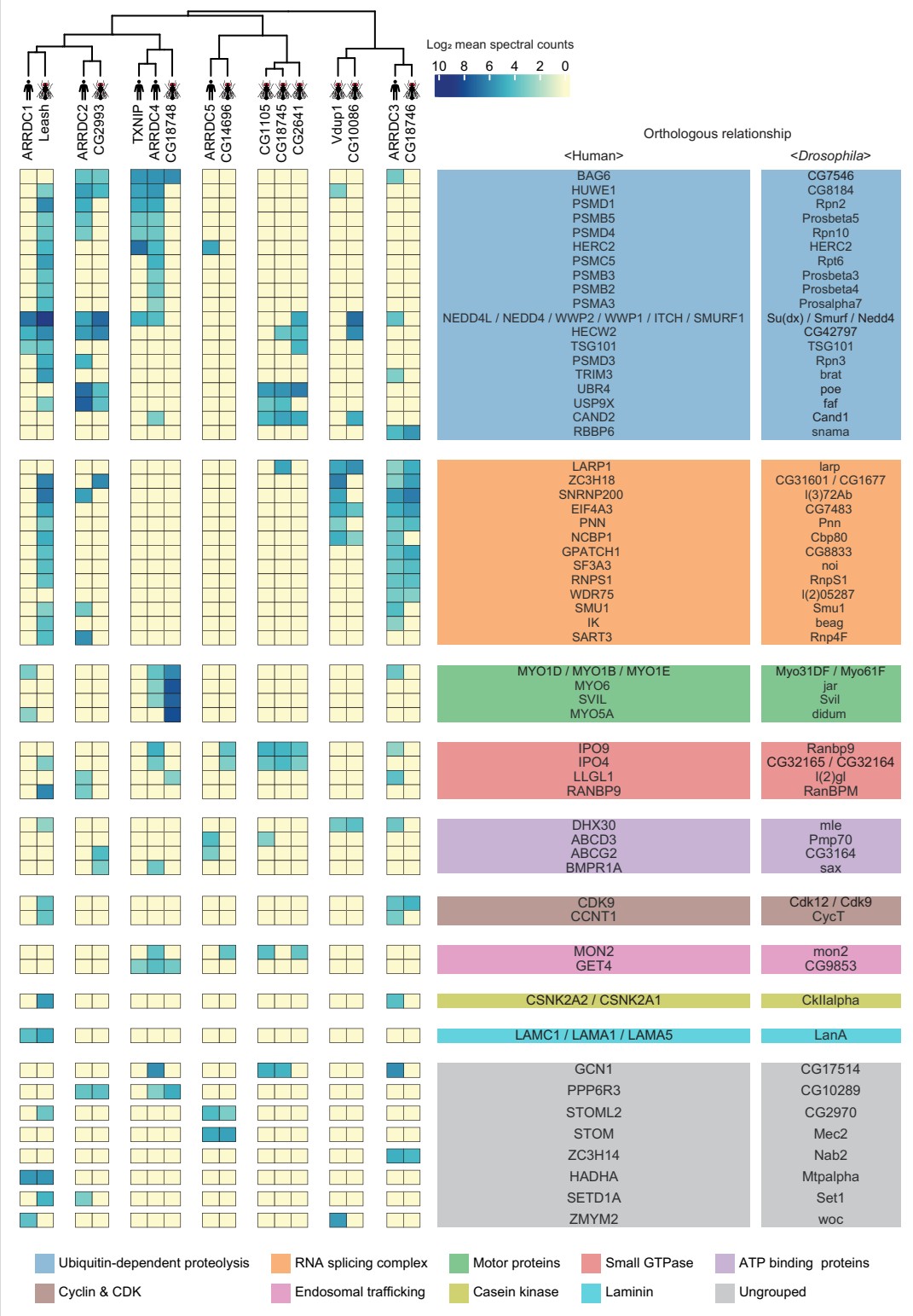

**Figure 4.** A substantial fraction of α-arrestin-protein–protein interactions (PPIs) are conserved across species. Human and *Drosophila* α-arrestins and their orthologous interactomes are hierarchically clustered based on log₂-transformed mean spectral counts. They are then manually grouped based on their shared biological functions and assigned distinct colors. The names of orthologous proteins that interact with α-arrestins are displayed on the right side of the heatmap.

The online version of this article includes the following figure supplement(s) for figure 4:

**Figure supplement 1.** Protein sequence homology of α-arrestins from human and *Drosophila*.

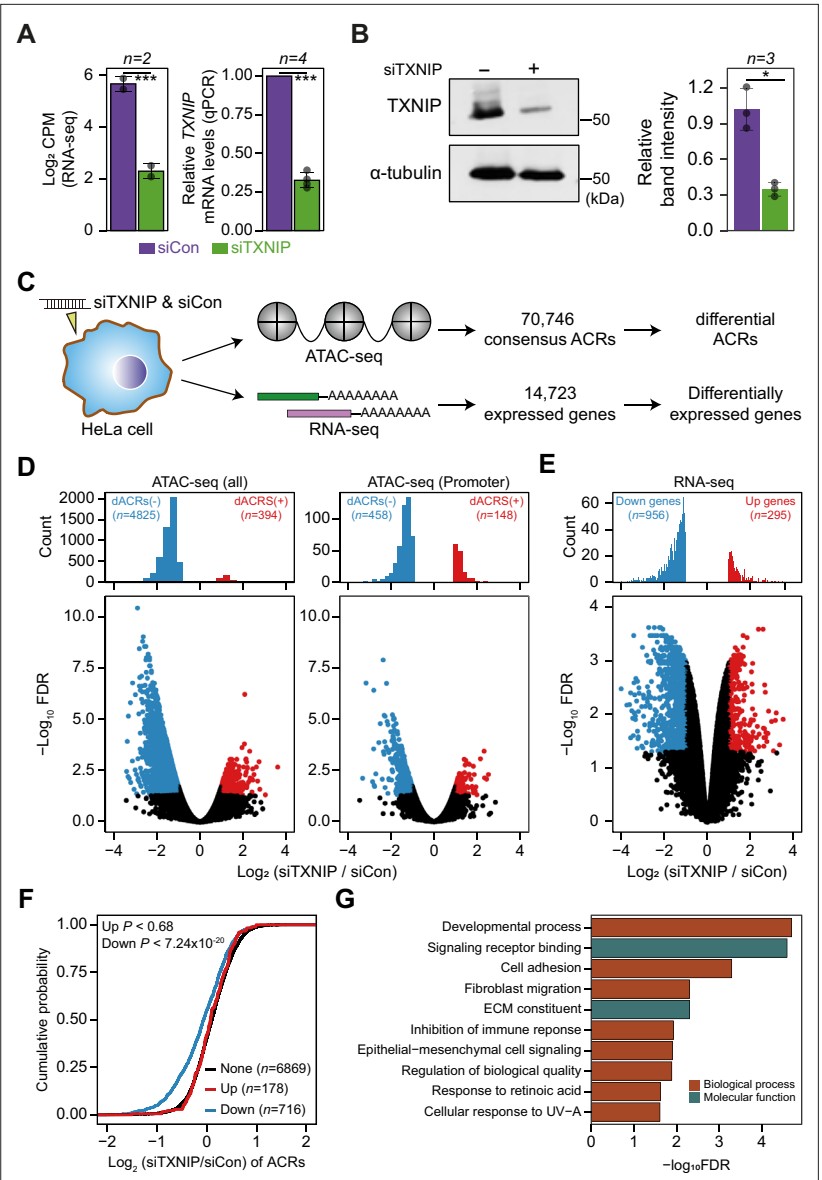

**Figure 5.** TXNIP knockdown induces a global decrease in chromatin accessibility and gene expression. (**A, B**) HeLa cells were treated with either siRNA against TXNIP (siTXNIP) or negative control (siCon) for 48 hr and analyzed of changes in the mRNA (**A**) and protein levels (**B**) of TXNIP. Gray dots depict actual values of each experiment and bar plots indicate mean ± SD. ***False discovery rate (FDR) < 0.001 (see 'Materials and methods') for RNA-seq. *p<0.05, ***p<0.001 (two-sided paired Student's t-test) for RT-qPCR and western blots. (**A**) Expression levels of RNAs were quantified by RNA-seq (left, log2 counts per million mapped reads [CPM], see 'Materials and methods') and RT-qPCR. (**B**) Protein levels were first visualized by western blot analysis of lysates from HeLa cells and band intensities of three independent experiments were quantified using imageJ software (right). (**C**) A schematic workflow for detecting differentially accessible chromatin regions (dACRs) and differentially expressed genes (DEGs) using ATAC- and RNA-seq analyses, respectively. (**D**) Volcano plots of differential chromatin accessibility for all accessible chromatin regions (ACRs) (left) and those associated with promoters (right). (**E**) Volcano plots of differential gene expression. (**D, E**) Blue dots denote 'dACRs(-)' of significantly decreased chromatin accessibility (**D**) and 'Down' genes of significantly downregulated genes (**E**) in siTXNIP-treated cells compared to control (FDR ≤ 0.05, log2(siTXNIP/siCon) ≤–1); red dots denote 'dACRs(+)' of significantly increased chromatin accessibility (**D**) and 'Up' genes of significantly upregulated genes (**E**) in siTXNIP-treated cells compared to control (FDR ≤ 0.05, log2(siTXNIP/siCon) ≥ 1). Black dots denote data points with no significant changes. (**F**) Changes in chromatin accessibility of ACRs located in the promoter region of genes were plotted as cumulative distribution functions (CDFs). Genes were categorized into three groups based on changes in RNA levels ('Up', 'Down' as in E and

*Figure 5 continued on next page*

*Figure 5 continued*

'None' indicating genes with –0.5 ≤log2(siTXNIP/siCon) ≤ 0.5). The number of genes in each group are shown in parentheses, and p-values in the left upper corner were calculated by one-sided Kolmogorov–Smirnov (KS) test. (**G**) Top 10 GO terms (biological process and molecular function) enriched in genes that exhibited decreased chromatin accessibility at their promoter and decreased RNA expression upon TXNIP knockdown (***Supplementary file 11***).

The online version of this article includes the following figure supplement(s) for figure 5:

**Figure supplement 1.** High-throughput sequencing data are highly reproducible, and ATAC-seq reads exhibit a typical pattern of strong enrichment around transcription start sites (TSSs) of expressed genes.

**Figure supplement 2.** Genomic locations of accessible chromatin regions (ACRs) and association between chromatin accessibility and transcriptional activity.

and promoter regions (29.23%, *Figure 5—figure supplement 2A*). TXNIP knockdown appeared to induce a global decrease in chromatin accessibility in many genomic regions including promoters (*Figure 5D*). Of the high-confidence ACRs, 7.38% were dACRs under TXNIP depletion; most dACRs showed reduced chromatin accessibility under this condition (dACRs(-), *Figure 5D*, *Figure 5—figure supplement 2B*). dACRs(-) were preferentially localized in gene bodies, whereas dACRs(+) were more often observed in promoter regions (*Figure 5—figure supplement 2C*).

The global chromatin changes induced by TXNIP knockdown could impact gene expression at corresponding loci. In fact, our gene expression analysis showed that 956 genes were downregulated and 295 genes were upregulated by TXNIP knockdown compared to the control (*Figure 5E*), suggesting that the global decrease in chromatin accessibility induced by TXNIP depletion would mediate the repression of gene expression. To confirm this phenomenon, we first selected sets of differentially ('Up' and 'Down' in *Figure 5F*) and non-DEGs ('None' in *Figure 5F*) with at least one detectable ACR in promoter or gene body. Next, the cumulative distribution function (CDF) of changes in chromatin accessibilities demonstrated that the genes with decreased RNA level ('Down') showed significantly reduced chromatin accessibilities at promoters compared to those with no changes in the RNA level ('None') (*Figure 5F*; $p<7.24 \times 10^{-20}$ for max changes; *Figure 5—figure supplement 2D*; $p<2.60 \times 10^{-24}$ for mean changes, Kolmogorov–Smirnov [KS] test). In contrast, genes with increased RNA expression ('Up') exhibited no changes in chromatin accessibilities at the promoter (*Figure 5F*; $p<0.68$ for max changes; *Figure 5—figure supplement 2D*; $p<0.49$ for mean changes, KS test), indicating that chromatin opening at promoters is necessary but not sufficient to induce gene expression. ACRs located in gene bodies also showed a similar trend: genes with a decreased RNA level ('Down') showing decreased chromatin accessibility upon TXNIP depletion (*Figure 5—figure supplement 2E*; $p<9.3 \times 10^{-4}$ for max changes and $p<2.58 \times 10^{-7}$ for mean changes, KS test), suggesting that TXNIP is likely to be a negative regulator of chromatin repressors that induce heterochromatin formation. We then used GO analysis (*Raudvere et al., 2019*) to examine the biological functions of genes that exhibited decreased chromatin accessibility at their promoter and decreased RNA expression upon TXNIP knockdown (***Supplementary file 11***). In general, genes associated with developmental process, signaling receptor binding, cell adhesion and migration, immune response, and extracellular matrix constituents appeared to be repressed upon TXNIP depletion (*Figure 5G*).

## TXNIP represses the recruitment of HDAC2 to target loci

Given that TXNIP knockdown led to a global reduction in chromatin accessibility with decreased transcription, we focused on identifying the potential role of the epigenetic silencer HDAC2, one of the strong binding partners of TXNIP in the AP/MS analysis, in mediating the TXNIP-dependent epigenetic and transcriptional modulation. Consistent with the AP/MS data, immunoprecipitation (IP) experiments showed that the two proteins indeed interact with each other. Furthermore, TXNIP knockdown reduced the amount of TXNIP-interacting HDAC2 protein but did not affect the HDAC2 expression level (*Figure 6A*). To find out how the TXNIP-HDAC2 interaction impacts the epigenetic and transcriptional reprogramming of target loci, we first checked whether the TXNIP-HDAC2 interaction causes cytosolic retention of HDAC2 to inhibit nuclear HDAC2-mediated global histone deacetylation. However, both the expression level and subcellular localization of HDAC2 were unaffected by a reduction in TXNIP, as confirmed by western blot analysis using cytoplasmic and nuclear fractions

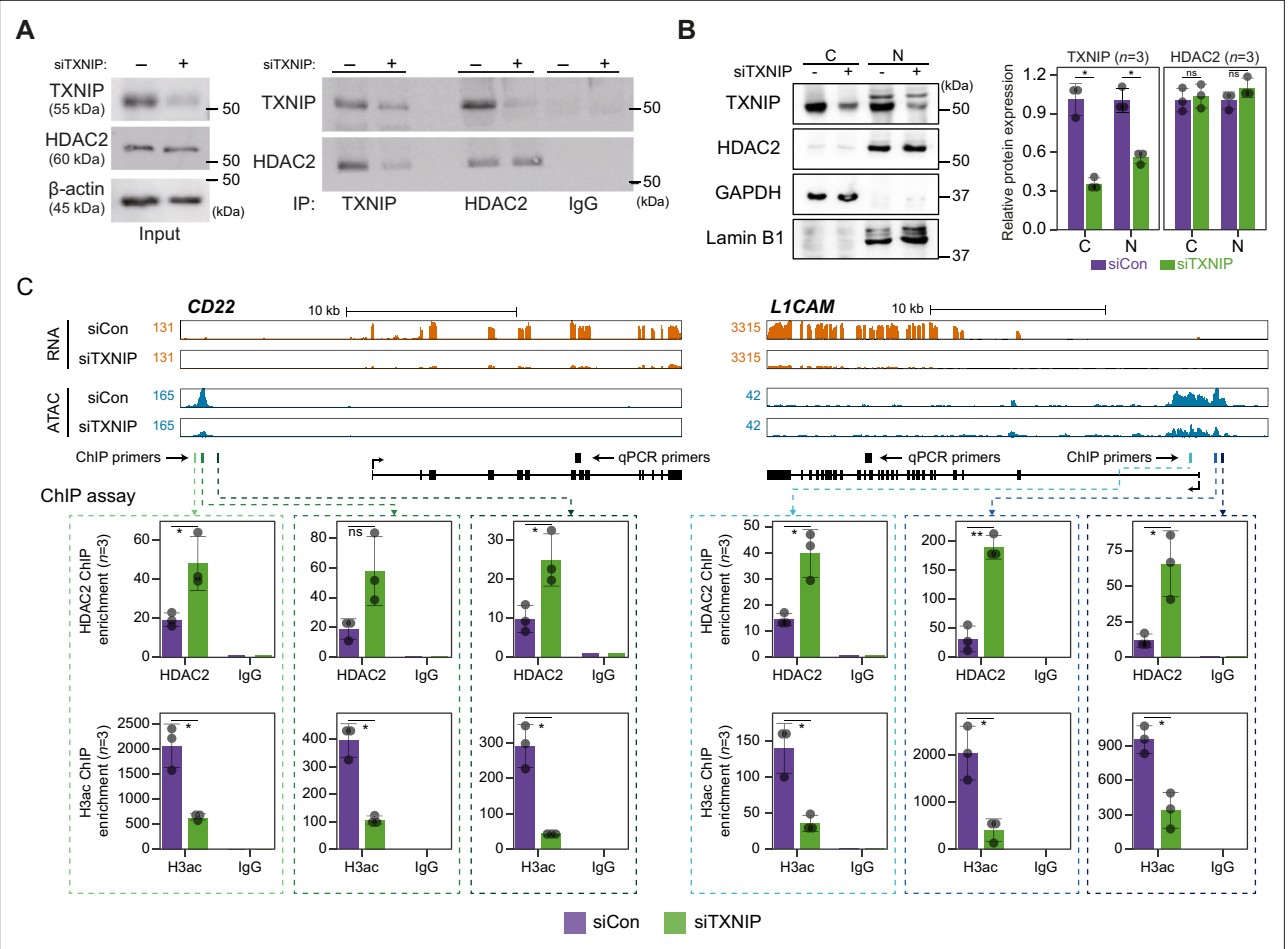

**Figure 6.** TXNIP directly represses the recruitment of HDAC2 to target loci. (**A**) Co-immunoprecipitation (Co-IP) assay showing the interaction between TXNIP and HDAC2 proteins. Lysates from HeLa cells that had been treated with either siCon or siTXNIP for 48 hr were subjected to IP and immunoblotting with antibodies recognizing TXNIP and HDAC2. IgG was used as the negative control. (**B**) Nuclear and cytoplasmic fractions of HeLa cells were analyzed with western blots following transfection with siCon or siTXNIP for 48 hr (left). Lamin B1 and GAPDH were used as nuclear and cytoplasmic markers, respectively. Western blot results from three independent experiments for TXNIP and HDAC2 were quantified and presented as in *Figure 5B*. C, cytoplasm; N, nucleus. (**C**) Genomic regions showing RNA expression and chromatin accessibility at CD22 and L1CAM gene loci (top). Through the ChIP-qPCR analysis, the fold enrichment of HDAC2 and histone H3 acetylation (H3ac) at the CD22 and L1CAM promoter regions in HeLa cells treated with either siCon or siTXNIP for 48 hr were quantified (bottom). Data are presented as the mean ± SD (n = 3, biological replicates). Gray dots depict actual values of each experiment. *p<0.05, **p<0.01, ns: not significant (two-sided paired Student's *t*-test).

The online version of this article includes the following figure supplement(s) for figure 6:

**Figure supplement 1.** TXNIP might play role in transcriptional regulation independent of known factors.

as well as by an immunofluorescence assay (*Figure 6B*, *Figure 6—figure supplement 1A*), indicating that TXNIP might modulate HDAC2 activity in a different way.

We next asked if the transcriptional suppression of TXNIP-target genes was mediated by changes in HDAC2 recruitment to and histone acetylation of chromatin. To address this question, genes that were significantly downregulated by TXNIP knockdown and that contained at least one dACR in the promoter were selected by the following additional criteria: (1) the RNA level in normal HeLa cells is ≥10 TPM and (2) the total ATAC-seq read count at the promoter in siTXNIP-treated HeLa cells is reduced ≥1.5-fold compared to that in normal cells. Among the four TXNIP-target genes selected by the above-mentioned criteria, the expression levels of CD22 and L1CAM were significantly reduced (p<0.05, Student's *t*-test, *Figure 6—figure supplement 1B*). The two genes were further examined to determine whether the levels of HDAC2-binding signal and histone acetylation in their promoter regions were changed upon TXNIP knockdown (*Figure 6C*). We observed that RNA- and ATAC-seq coverages in exonic and promoter region of CD22 and L1CAM genes were clearly reduced upon

TXNIP depletion (*Figure 6C*, top), and an analysis of chromatin immunoprecipitation (ChIP) signals for HDAC2 and histone H3 acetylation at each dACR(-) detected in the L1CAM and CD22 promoters revealed that TXNIP knockdown increased the recruitment of HDAC2 to TXNIP-target loci, accompanied by decreased histone H3 acetylation (*Figure 6C*, bottom). Therefore, these results suggest that the TXNIP interaction with HDAC2 inhibits the chromatin occupancy of HDAC2 and subsequently reduces histone deacetylation to facilitate global chromatin accessibility.

HDAC2 typically operates within the mammalian nucleus as part of co-repressor complexes as it lacks the ability to bind to DNA directly (*Hassig et al., 1997*). The nucleosome remodeling and deacetylation (NuRD) complex is one of the well-recognized co-repressor complexes that contains HDAC2 (*Kelly and Cowley, 2013*; *Seto and Yoshida, 2014*), and we sought to determine if depletion of TXNIP affects interaction between HDAC2 and other components in this NuRD complex. While HDAC2 interacted with MBD3 and MTA1 under normal condition, the interaction between HDAC2 and MBD3 or MTA1 was not affected upon TXNIP depletion (*Figure 6—figure supplement 1C*). Next, given that HDAC2 phosphorylation is known to influence its enzymatic activity and stability (*Adenuga and Rahman, 2010*; *Adenuga et al., 2009*; *Bahl and Seto, 2021*; *Tsai and Seto, 2002*), we tested if TXNIP depletion alters phosphorylation status of HDAC2. The result indicated, however, that phosphorylation status of HDAC2 does not change upon TXNIP depletion (*Figure 6—figure supplement 1D*). In summary, our findings suggest a model where TXNIP plays a role in transcriptional regulation independent of these factors (*Figure 6—figure supplement 1E*). When TXNIP is present, it directly interacts with HDAC2, a key component of transcriptional co-repressor complex. This interaction suppresses the HDAC2's recruitment to target genomic regions, leading to the histone acetylation of target loci possibly through active complex including histone acetyltransferase (HAT). As a result, transcriptional activation of target gene occurs. In contrast, when TXNIP expression is diminished, the interaction between TXNIP and HDAC2 weakens. This restores histone deacetylating activity of HDAC2 in the co-repressor complex, leading to subsequent repression of target gene transcription.

## ARRDC5 plays a role in osteoclast differentiation and function

Given that various subunits of the V-type ATPase interact with ARRDC5, we speculated that ARRDC5 might be involved in the function of this complex (*Figure 7A*). V-type ATPase plays an important role in the differentiation and function of osteoclasts, which are multinucleated cells responsible for bone resorption in mammals (*Feng et al., 2009*; *Qin et al., 2012*). Therefore, we hypothesized that ARRDC5 might be also important for osteoclast differentiation and function. To determine whether ARRDC5 affects osteoclast function, we prepared osteoclasts by infecting bone marrow-derived macrophages (BMMs) with lentivirus expressing either GFP-GFP or GFP-ARRDC5 and differentiating the cells into mature osteoclasts. After 5 days of differentiation, ectopic expression of GFP-ARRDC5 had significantly increased the total number of tartrate-resistant acid phosphatase (TRAP)-positive multinucleated cells compared to GFP-GFP overexpression (*Figure 7B*). In particular, the number of TRAP-positive osteoclasts with a diameter larger than 200 μm was significantly increased by GFP-ARRDC5 overexpression (*Figure 7B*), suggesting that ARRDC5 expression increased osteoclast differentiation. Additionally, the area of resorption pits produced by GFP-ARRDC5-expressing osteoclasts in a bone resorption pit assay was approximately fourfold greater than that of GFP-GFP-expressing osteoclasts (*Figure 7C*). Depletion of ARRDC5 using short hairpin RNA (shRNA) impaired osteoclast differentiation, further affirming its crucial role in this differentiation process (*Figure 7D and E*). These results imply that ARRDC5 promotes osteoclast differentiation and bone resorption activity.

The V-type ATPase is localized at the osteoclast plasma membrane (*Toyomura et al., 2003*), and its localization is important for cell fusion, maturation, and function during osteoclast differentiation (*Feng et al., 2009*; *Qin et al., 2012*). Furthermore, its localization is disrupted by bafilomycin A1, which is shown to attenuate the transport of the V-type ATPase to the membrane (*Matsumoto and Nakanishi-Matsui, 2019*). We analyzed changes in the expression level and localization of V-type ATPase, especially V-type ATPase V1 domain subunit (ATP6V1), in GFP-GFP and GFP-ARRDC5-overexpressing osteoclasts. The level of V-type ATPase expression did not change in osteoclasts regardless of ARRDC5 expression levels (*Figure 7F*). GFP signals were detected at the cell membrane when GFP-ARRDC5 was overexpressed, indicating that ARRDC5 might also localize to the osteoclast plasma membrane (*Figure 7G*). In addition, we detected more V-type ATPase signals at the cell membrane in the GFP-ARRDC5-overexpressing osteoclasts, and ARRDC5 and V-type ATPase were

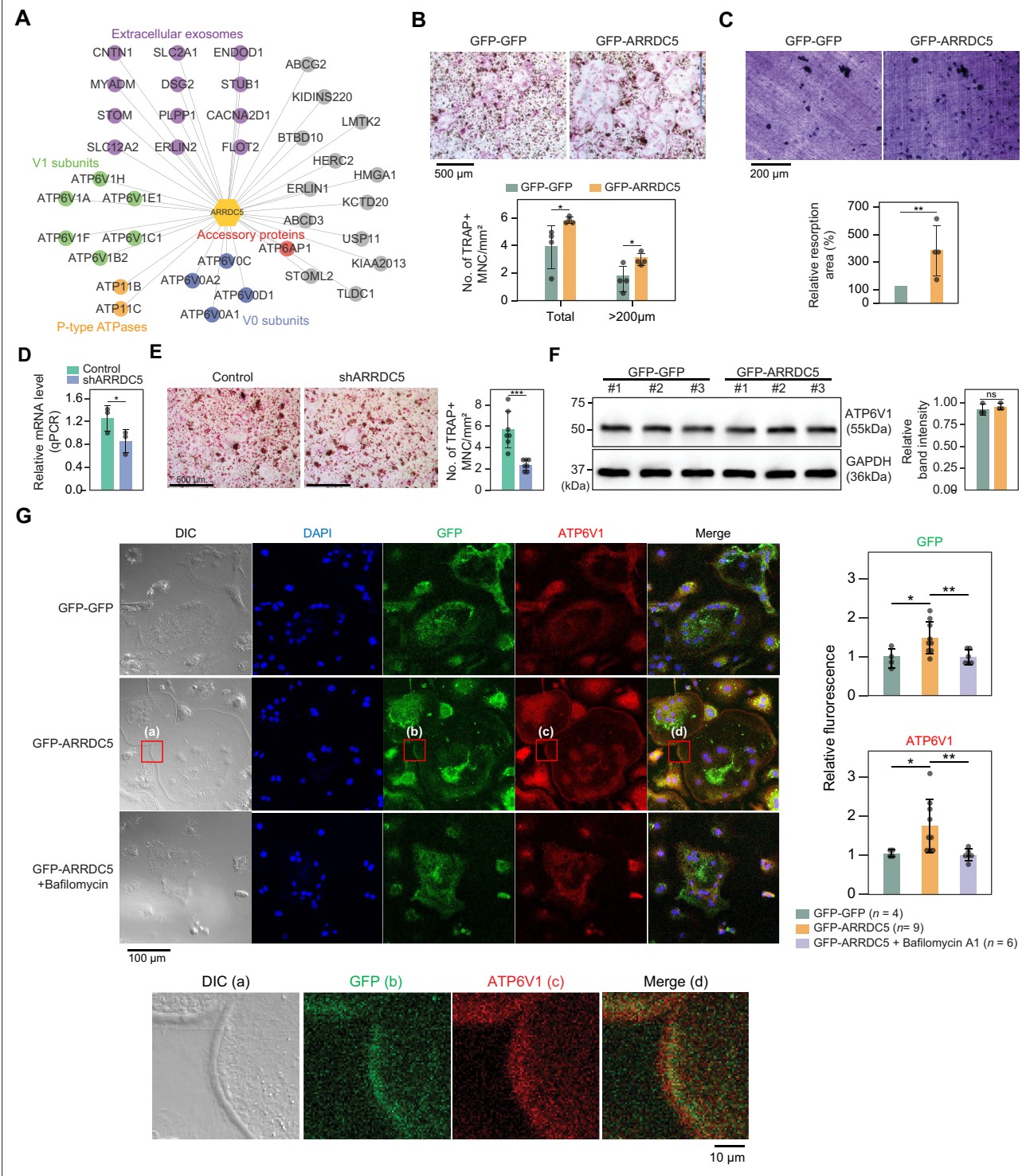

**Figure 7.** Interaction of ARRDC5 with the V-type ATPases in osteoclasts. (**A**) The human ARRDC5-centric protein–protein interactions (PPI) network. V-type and P-type ATPases, their related components, and extracellular exosomes are labeled and colored. Other interacting proteins are indicated with gray circles. (**B**) TRAP staining of osteoclasts. Cell differentiation was visualized with TRAP staining of GFP-GFP or GFP-ARRDC5-overexpressing osteoclasts (scale bar = 500 μm). TRAP-positive multinucleated cells (TRAP + MNC) were quantified as the total number of cells and the number of cells whose diameters were greater than 200 μm. *p<0.05. Data are presented as the mean ± SD, n=4. (**C**) Resorption pit formation on dentin slices. Cell activity was determined by measuring the level of resorption pit formation in GFP-GFP or GFP-ARRDC5-overexpressing osteoclasts (scale bar = 200 μm). Resorption pits were quantified as the percentage of resorbed bone area per the total dentin disc area using ImageJ software. The resorption area is relative to that in dentin discs seeded with GFP-GFP-overexpressing osteoclasts, which was set to 100%. The colors of the bar plots are same as in (**B**).

*Figure 7 continued on next page*

*Figure 7 continued*

**p<0.01. Data are presented as the mean ± SD, $n$ = 4. (**D**) Relative mRNA levels of ARRDC5 in non-target control (Control) or shARRDC5-expressing osteoclasts (shARRDC5) measured by qPCR. *p<0.05 (Student's $t$-test, one-sided). Data are presented as the mean ± SD, $n$=3. (**E**) TRAP staining of osteoclasts. Cell differentiation was visualized with TRAP staining of 'Control' or 'shARRDC5'-expressing osteoclasts (scale bar = 500 µm, left). TRAP-positive multinucleated cells (TRAP + MNC) were quantified as the total number of cells (right). Colors of the bar plots are same as in (**D**). ***p<0.001 (Student's $t$-test). Data are presented as the mean ± SD, $n$=7. (**F**) The protein level of ATP6V1 in GFP-GFP or GFP-ARRDC5-overexpressing osteoclasts. The numbers represent independent samples for western blot analysis (left) and band intensities of three independent experiments were quantified (right). Colors of the bar plots are same as in (**B**) and (**C**). ns, not significant. Data are presented as the mean ± SD, $n$=3. (**G**) Localization of ARRDC5 and V-type ATPase V1 domain subunit (ATP6V1) in osteoclasts. ATP6V1 was visualized with immunofluorescence (red), GFP-GFP and GFP-ARRDC5 were visualized with GFP fluorescence (green), and nuclei were visualized with DAPI (blue). Representative fluorescence images are shown (scale bar = 100 µm). The region of interest, marked by the red boxes, was high-magnified and presented below (scale bar = 10 µm). The integrated density of fluorescence was quantified using ImageJ software and expressed as relative fluorescence (right). The integrated density of fluorescence in GFP-GFP osteoclasts was established as the reference value, which was set to 1. *p<0.05, **p<0.01 (Student's $t$-test). Data are presented as the mean ± SD with the number of data indicated in the figure.

co-localized at the osteoclast membrane (*Figure 7G*). Notably, bafilomycin A1 treatment reduced not only the V-type ATPase signals detected at the cell membrane but also the GFP-ARRDC5 signals. These results indicate that ARRDC5 might control the plasma membrane localization of the V-type ATPase during osteoclast differentiation and function.

## Discussion

In this study, we constructed high-confidence interactomes of α-arrestins from human and *Drosophila*, comprising 307 and 467 interacting proteins, respectively. The resulting interactomes greatly expanded previously known PPIs involving α-arrestins and the majority of interactomes were first reported in this study, which needs to be validated experimentally. However, some known PPIs were missed in our interactomes due to low spectral counts and SAINTexpress scores, probably resulting from different cellular contexts, experimental conditions, or other factors (*Figure 1—figure supplement 2F*). According to a phylogenetic analysis of arrestin family proteins, α-arrestins were shown to be ubiquitously conserved from yeast to human (*Alvarez, 2008*). However, compared to the more established visual/β-arrestin proteins, α-arrestins have been discovered more recently and much of their molecular mechanisms and functions remain mostly unexplored except for budding yeast model (*Zbieralski and Wawrzycka, 2022*). Based on the high-confidence interactomes of α-arrestins from human and *Drosophila*, we identified conserved and specific functions of these α-arrestins. Furthermore, we uncovered molecular functions of newly discovered function of human specific α-arrestins, TXNIP and ARRDC5. We anticipate that the discovery made here will enhance current understanding of α-arrestins.

Integrative map of protein complexes that interact with α-arrestins (*Figures 2 and 3*, *Figure 2—figure supplement 1*, *Figure 3—figure supplement 1*) hint toward many aspects of α-arrestins's biology that remain uncharacterized. For example, the role of α-arrestins in the regulation of β2AR in human remained controversial. One study proposed that α-arrestins might act coordinately with β-arrestins at the early step of endocytosis, promoting ubiquitination, internalization, endosomal sorting, and lysosomal degradation of activated GPCRs (*Shea et al., 2012*). Another study, however, proposed a different hypothesis, suggesting that α-arrestins might act as a secondary adaptor localized at endosomes to mediate endosomal sorting of cargo molecules (*Han et al., 2013*). Among the protein complexes that interact with α-arrestins, we identified those related with clathrin-coated pit in human (*Figure 2*, *Figure 2—figure supplement 1*) and AP-2 adaptor complex in *Drosophila* (*Figure 3*, *Figure 3—figure supplement 1*). They are multimeric proteins to induce internalization of cargo molecules to mediate clathrin-mediated endocytosis, which suggests involvement of α-arrestins in early step of endocytosis.

The integrative map of protein complexes also highlighted both conserved and unique relationships between α-arrestins and diverse functional protein complexes. For instance, protein complexes involved in ubiquitination-dependent proteolysis, proteasome, RNA splicing, and intracellular transport (motor proteins) were prevalently linked with α-arrestins in both human and *Drosophila*. To more precisely identify conserved PPIs associated with α-arrestins, we undertook ortholog predictions within the α-arrestins' interactomes. This revealed 58 orthologous interaction groups that were

observed to be conserved between human and *Drosophila* (*Figure 4*). Among conserved proteins, proteins known to interact with human α-arrestins, such as NEDD4, WWP2, WWP1, and ITCH, were identified along with its orthologs in *Drosophila*, which are Su(dx), Nedd4, and Smurf, implying that regulatory pathway of ubiquitination-dependent proteolysis by α-arrestins is also present in invertebrate species. Besides the known conserved functions, the novel conserved functions of α-arrestins interactomes were also identified, such as RNA splicing (*Figure 4*). Because our protocol did not include treatment with RNase before the AP/MS, it is possible that RNA-binding proteins could co-precipitate with other proteins that directly bind to α-arrestins through RNAs, and thus could be indirect binding partners. Nevertheless, other RNA-binding proteins except for RNA splicing and processing factors were not enriched in our interactomes, indicating that this possibility may be not the case. Thus, it might be of interest to explore how α-arrestins are linked to RNA processing in future. Additionally, interaction between α-arrestins and entities like motor proteins, small GTPase, ATP-binding proteins, and endosomal trafficking components was identified to be conserved. Further validation of these interactions could unveil molecular mechanisms consistently associated with these cellular functions.

Some protein complexes and functional modules were found to be involved in specific cellular processes discovered in only human, suggesting that some functional roles of α-arrestins have diverged through evolution. As examples of specific cellular functions of α-arrestins, we explored the biological relevance of two interacting protein complexes: (1) the interaction between TXNIP and chromatin remodelers and (2) the interaction between ARRDC5 and the V-type ATPase complex. Given that TXNIP interacts with chromatin remodelers, such as the HDAC, we speculated that chromatin structures could be affected by the interactions. Although we showed that siTXNIP treatment directed a global decrease in chromatin accessibilities and gene expression by inhibiting the binding of HDAC2 to targets, histones themselves could be also controlled by the interaction between TXNIP and the H2B ubiquitination complex. An impact of TXNIP on histone ubiquitination could strengthen the negative regulation of target loci by siTXNIP treatment. In addition, TXNIP interacts with the proteasome, which induces the degradation of binding partners (*Figure 2*, *Figure 2—figure supplement 1*). However, we observed that the cellular level and localization of HDAC2 were not affected by TXNIP reduction (*Figure 6A and B*, *Figure 6—figure supplement 1A*), meaning that the proteasome seems not to be involved in TXNIP's influence on HDAC2; rather, TXNIP directly hinders HDAC2 recruitment to target loci.

Because the V-type ATPase plays a key role in osteoclast differentiation and physiology (*Feng et al., 2009*; *Qin et al., 2012*), we investigated a possible role of the ARRDC5-V-type ATPase interaction in this cell type. We demonstrated that the ectopic expression of ARRDC5 enhances both the differentiation of osteoclasts into their mature form and their bone reabsorption activity (*Figure 7B and C*). Conversely, depletion of ARRDC5 reduces osteoclast maturation, underscoring the pivotal role of ARRDC5 in osteoclast development and function (*Figure 7D and E*). Additionally, ARRDC5 co-localized with the V-type ATPase at the plasma membrane (*Figure 7G*). Thus, further characterization of ARRDC5 and its interactome in osteoclasts might clarify how ARRDC5 regulates the V-type ATPase to play a role in osteoclast differentiation and function. With the results, the discovery of new binding partners and their functions of TXNIP and ARRDC5 will facilitate further investigations to explore the novel PPIs of α-arrestins.

Given the plethora of PPIs uncovered in this study, we also anticipate that our study could provide insight into many disease models. In fact, despite a limited knowledge of their biology, α-arrestins have already been linked to a range of cellular processes and several major health disorders, such as diabetes (*Batista et al., 2020*; *Wondafrash et al., 2020*), cardiovascular diseases (*Domingues et al., 2021*), neurological disorders (*Tsubaki et al., 2020*), and tumor progression (*Chen et al., 2020*; *Mohankumar et al., 2015*; *Oka et al., 2006*), making them potential therapeutic targets. We further explored association between α-arrestins' interactomes and disease pathways (*Figure 8*). Notably, the interactomes of α-arrestins in human showed clear links to specific diseases. For instance, ARRDC5 is closely associated with diseases resulting from viral infection and cardiovascular conditions. ARRDC2, ARRDC4, and TXNIP share common association with certain neurodegenerative diseases, while ARRDC1 is implicated in cancer.

Lastly, to assist the research community, we have made comprehensive α-arrestin interactome maps on our website (big.hanyang.ac.kr/alphaArrestin_PPIN). Researchers can search and download

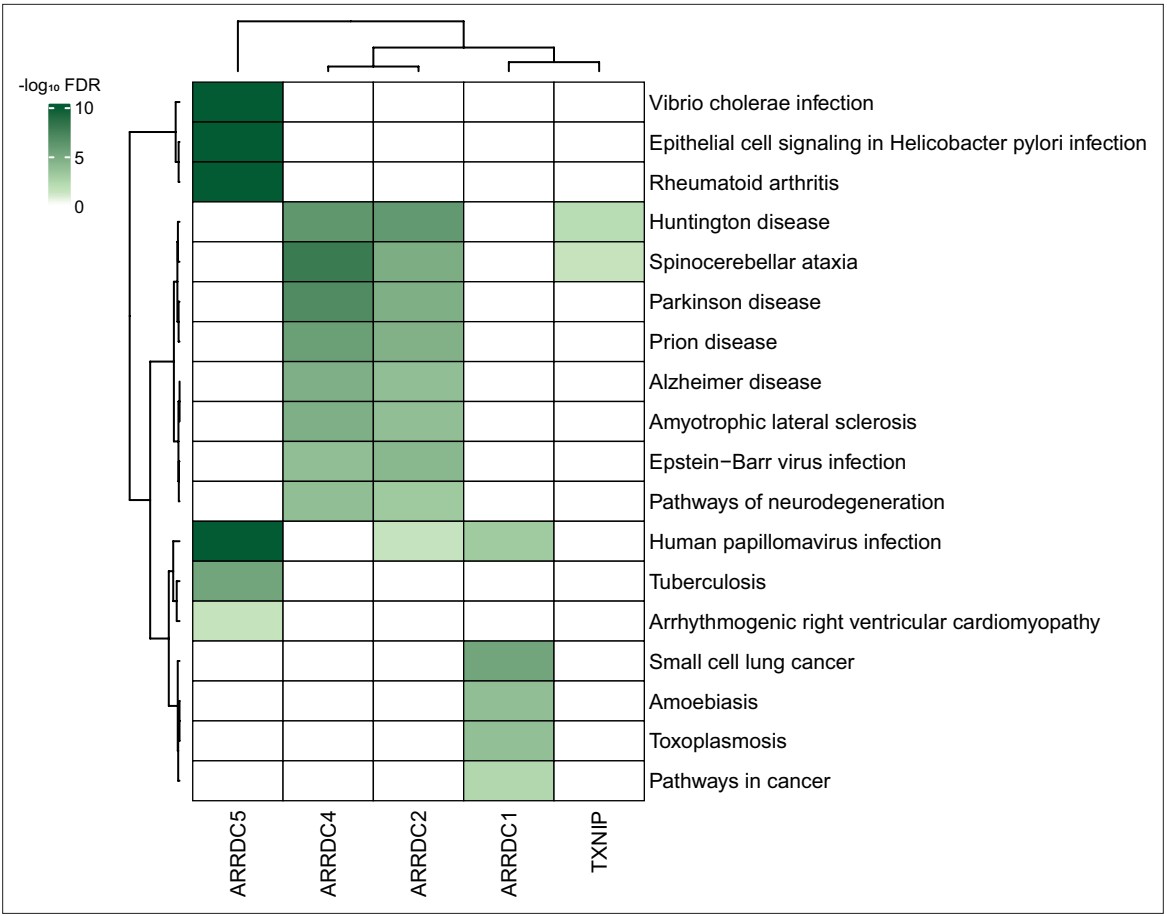

**Figure 8.** Association between α-arrestin interactomes and human diseases. Heatmap depicts disease pathways from the Kyoto Encyclopedia of Genes and Genomes (KEGG) that are enriched in interactome of each α-arrestin. The significance of the enrichment was tested by enrichR (**Kuleshov et al., 2016**) and indicated as -log$_{10}$ FDR. Only the disease pathways that are significantly enriched (FDR < 0.05) are colored. FDR, false discovery rate.

their interactomes of interest as well as access information on potential cellular functions and protein class associated with these interactomes.

# Materials and methods

## Key resources table

| Reagent type (species) or resource | Designation | Source or reference | Identifiers | Additional information |
|---|---|---|---|---|
| Strain, strain background (*Mus musculus*) | Bone marrow-derived macrophages (BMMs) | KOATECH (Gyeonggi-do, South Korea) | KOATECH:C5BL/6 | |
| Genetic reagent (*M. musculus*) | Arrdc5 shRNA | This paper | | pLKO.1-puro-CMV-tGFP vector (SHC003; Sigma Aldrich) containing target sequence 5'-CCACACCTTTGAACTTCCATTT-3' |
| Cell line (*Homo sapiens*) | HeLa | American Type Culture Collection (ATCC) | ATCC:CCL-2 | |
| Cell line (*H. sapiens*) | HEK293 | American Type Culture Collection (ATCC) | ATCC:CRL-1573 | |

*Continued on next page*

*Continued*

| Reagent type (species) or resource | Designation | Source or reference | Identifiers | Additional information |
|---|---|---|---|---|
| Cell line (*H. sapiens*) | HEK293T | American Type Culture Collection (ATCC) | ATCC:CRL-3216 | |
| Cell line (*Drosophila melanogaster*) | S2R+ | *Drosophila* Genomics Resource Center (DGRC) | DGRC:Stock number 150 | |
| Antibody | TXNIP (D5F3E) Rabbit mAb | Cell Signaling Technology | Cell signaling:14715 | |
| Antibody | HDAC2 (D6S5P) Rabbit mAb | Cell Signaling Technology | Cell signaling:57156 | |
| Antibody | Histone H3ac (pan-acetyl) antibody (pAb) 100 µl | Active Motif | Active Motif:39139 | |
| Antibody | normal rabbit IgG | Santa Cruz Biotechnology | Santa Cruz:sc-2027 | |
| Antibody | Rabbit TrueBlot: Anti-Rabbit IgG HRP | RockLand | RockLand:18-8816-31 | |
| Antibody | Monoclonal Anti-ATP6V1A, (C-terminal) antibody produced in mouse, clone 4 F5, purified immunoglobulin, buffered aqueous solution | Sigma Aldrich | Sigma Aldrich:SAB1402125-100UG | |
| Antibody | Goat anti-Mouse IgM (Heavy chain) Cross-Adsorbed Secondary Antibody, Alexa Fluor 594 | Invitrogen | Invitrogen:A-21044 | |
| Antibody | Rabbit Anti-Mouse IgG H&L (HRP) | Abcam | Abcam:ab6728 | |
| Antibody | Goat Anti-Rabbit IgG H&L (HRP) | Abcam | Abcam:ab6721 | |
| Antibody | α-Tubulin (DM1A) Mouse mAb | Cell Signaling Technology | Cell Signaling:3873 | |
| Antibody | Fluorescein (FITC) AffiniPure Donkey Anti-Rabbit IgG (H+L) | Jackson ImmunoResearch Laboratories | Jackson ImmunoResearch:711-095-152 | |
| Antibody | Cy3 AffiniPure Donkey Anti-Rabbit IgG (H+L) | Jackson ImmunoResearch Laboratories | Jackson ImmunoResearch:711-165-152 | |
| Antibody | HDAC2 Antibody | Cell Signaling Technology | Cell Signaling:2540 | |
| Antibody | GAPDH (D16H11) XP Rabbit mAb | Cell Signaling Technology | Cell Signaling:5174 | |
| Antibody | Lamin B1 (D9V6H) Rabbit mAb | Cell Signaling Technology | Cell Signaling:13435 | |
| Antibody | Phospho-HDAC2 (Ser394) (E8O2Z) Rabbit mAb | Cell Signaling Technology | Cell Signaling:69238 | |

*Continued on next page*

*Continued*

| Reagent type (species) or resource | Designation | Source or reference | Identifiers | Additional information |
|---|---|---|---|---|
| Antibody | MTA1 (D40D1) XP Rabbit mAb | Cell Signaling Technology | Cell Signaling:5647 | |
| Antibody | MBD3 (N87) Antibody | Cell Signaling Technology | Cell Signaling:14540 | |
| Antibody | ATP6V1B2 (D2F9R) Rabbit mAb | Cell Signaling Technology | Cell Signaling:14617 | |
| Antibody | Anti-rabbit IgG, HRP-linked Antibody | Cell Signaling Technology | Cell Signaling:7074 S | |
| Antibody | Anti-mouse IgG, HRP-linked Antibody | Cell Signaling Technology | Cell Signaling:7076 S | |
| Antibody | GAPDH (G-9) | Santa Cruz Biotechnology | Santa Cruz:sc-365062 | |
| Recombinant DNA reagent | pCR8/GW/TOPO TA cloning kit | Thermo Fisher Scientific | Thermo Fisher:K250020 | |
| Recombinant DNA reagent | pMK33-Gateway-GFP destination vector | *Kwon et al., 2013* | pMK33 | |
| Recombinant DNA reagent | pHAGE-GFP-Gateway destination vector | Other | | Gift from Dr. Chanhee Kang at Seoul National University |
| Recombinant DNA reagent | PEIPro DNA transfection reagent | VWR international | VWR:115010 | |
| Recombinant DNA reagent | Gateway LR Clonase II enzyme mix | Thermo Fisher Scientific | Thermo Fisher:11791020 | |
| Sequence-based reagent | α-tubulin RT-qPCR primers | This paper | | "Forward:CTGGACCGCATCTCTGTGTACT;Reverse:GCCAAAAGGACCTGAGCGAACA" |
| Sequence-based reagent | TXNIP RT-qPCR primers | This paper | | "Forward:GCTCCTCCCTGCTATATGGAT;Reverse:AGTATAAGTCGGTGGTGGCAT" |
| Sequence-based reagent | CD22 RT-qPCR primers | This paper | | "Forward:GCGCAGCTTGTAATAGTTGGTGC;Reverse:CACATTGGAGGCTGACCGAGTT" |
| Sequence-based reagent | L1CAM RT-qPCR primers | This paper | | "Forward:TCGCCCTATGTCCACTACACCT;Reverse:ATCCACAGGGTTCTTCTCTGGG" |
| Sequence-based reagent | OTULINL RT-qPCR primers | This paper | | "Forward:GTGTGGAGGCAGAGGTTGAT;Reverse:ATGCCGCCAAAATAGCTCCT" |
| Sequence-based reagent | PRR5L RT-qPCR primers | This paper | | "Forward:GCGGCTGTTGAAGAGTGAAC;Reverse:AGCCAGAACCTCAATGCGAT" |
| Sequence-based reagent | SDC3 RT-qPCR primers | This paper | | "Forward:CTCCTGGACAATGCCATCGACT;Reverse:TGAGCAGTGTGACCAAGAAGGC" |
| Sequence-based reagent | GAPDH1 RT-qPCR primers | This paper | | "Forward:ATCACCATCTTCCAGGAGCGA;Reverse:CCTTCTCCATGGTGGTGAAGAC" |
| Sequence-based reagent | CD22 ChIP-qPCR primers | This paper | | "Forward#1:CGCTGGAGAAGTGAGTTCGG;Reverse#1:TCCCTGCCTCCACTGATAGC", "Forward#2:GACGCTGAGATGAGGGTTGG;Reverse#2:TGACTCAGGAGGTTGGCAGA", "Forward#3:TCCCCACTCTTCTCGCTCTC;Reverse#3:ATTTGCGAGGTTGAGGTTGTC" |
| Sequence-based reagent | L1CAM ChIP-qPCR primers | This paper | | "Forward#1:CAGCTCAGTGCCTCATGGAA;Reverse#1:GAGACTGCTTCCAGAGTGGG", "Forward#2:GGAATGCTTCACTGGGCAAC;Reverse#2:GGGGTAAGAATTCCGGAGCC", "Forward#3:CGTGTCTGAGAAAGGAAGCCA;Reverse#3:CGGCTTATCCCGATCTACCC" |
| Sequence-based reagent | TXNIP siRNA | Bioneer (Dajeon, South Korea) | | "Sense: 5'-GUCAGUCACUCUCAGCCAUdTdT–3';Anti-sense: 5'-AUGGCUGAGAGUGACUGACdTdT-3'" |

*Continued on next page*

*Continued*

| Reagent type (species) or resource | Designation | Source or reference | Identifiers | Additional information |
|---|---|---|---|---|
| Sequence-based reagent | AccuTarget Negative control siRNA | Bioneer (Dajeon, South Korea) | | |
| Peptide, recombinant protein | Recombinant Human M-CSF | PeproTech | PeproTech:300–25 | |
| Peptide, recombinant protein | Recombinant Mouse TRANCE/RANK L/TNFSF11 | R&D Systems | R&D Systems:462-TEC | |
| Peptide, recombinant protein | Bafilomycin A1 | Sigma Aldrich | Sigma Aldrich:19–148 | |
| Commercial assay or kit | Pierce BCA Protein Assay Kit | Thermo Fisher Scientific | Thermo Fisher:23225 | |
| Commercial assay or kit | The ChIP-IT High Sensitivity (HS) Kit | Active Motif | Active Motif:53040 | |
| Commercial assay or kit | Effectene Transfection Reagent | Qiagen | Qiagen:301425 | |
| Commercial assay or kit | NE-PER Nuclear and Cytoplasmic Extraction Reagents | Thermo Fisher Scientific | Thermo Fisher:78833 | |
| Commercial assay or kit | Lipofectamine RNAiMAX | Invitrogen | Invitrogen:13778075 | |
| Commercial assay or kit | CRISPR & MISSION Lentiviral Packaging Mix | Sigma Aldrich | Sigma Aldrich:SHP002 | |
| Commercial assay or kit | TRAP Staining Kit | Cosmo Bio Co., LTD | Cosmo Bio:PMC-AK04F-COS | |
| Commercial assay or kit | dentin discs | Immunodiagnostic Systems (IDS) | IDS:AE-8050 | |
| Commercial assay or kit | ReverTra Ace qPCR RT Kit | Toyobo | Toyobo:FSQ-101 | |
| Commercial assay or kit | GoScript Reverse Transcriptase | Promega | Promega:A5001 | |
| Commercial assay or kit | TruSeq Stranded mRNA Sample Prep Kit | Illumina | Illumina:RS-122–2101 | |
| Commercial assay or kit | SuperScript II reverse transcriptase | Invitrogen | Invitrogen:18064014 | |
| Commercial assay or kit | Illumina Tagment DNA TDE1 Enzyme and Buffer Kits | Illumina | Illumina:20034197 | |
| Commercial assay or kit | Nextera DNA Flex kit | Illumina | Illumina#20018704 | |
| Commercial assay or kit | MinElute PCR purification Kit | Qiagen | Qiagen#28004 | |
| Commercial assay or kit | Mycoplasma PCR Detection Kit | abm | abm#G238 | |
| Commercial assay or kit | e-Myco plus Mycoplasma PCR Detecting Kit | iNtRON Biotechnology | iNtRON#25237 | |

*Continued*

| Reagent type (species) or resource | Designation | Source or reference | Identifiers | Additional information |
|---|---|---|---|---|
| Chemical compound, drug | Histopaque | Sigma Aldrich | Sigma Aldrich:1077 | |
| Software, algorithm | SAINTexpress | *Teo et al., 2014* | | Version 3.6.1 |
| Software, algorithm | COMPLEAT | *Vinayagam et al., 2013* | | |
| Software, algorithm | DAVID | *Huang et al., 2009a; Huang et al., 2009b* | | |
| Software, algorithm | DIOPT | *Hu et al., 2011* | | Version 7.1 |
| Software, algorithm | Cytoscape | *Shannon et al., 2003* | | Version 3.5.1 and 3.8.2 |
| Software, algorithm | ENCODE ATAC-seq pipeline | *Jin-Wook et al., 2018* | | Version 1.9.2 |
| Software, algorithm | FastQC | *Andrews, 2010* | | Version 0.11.8 |
| Software, algorithm | Sickle | *Joshi and Fass, 2011* | | Version 1.33 |
| Software, algorithm | STAR | *Dobin et al., 2013* | | Version 2.5.3a |
| Software, algorithm | RSEM | *Li and Dewey, 2011* | | Version 1.3.1 |
| Software, algorithm | Comet search engine | *Eng et al., 2013* | | |
| Software, algorithm | T-COFFEE | *Notredame et al., 2000* | | |
| Software, algorithm | RAxML | *Stamatakis, 2014* | | Version 8.2.11 |
| Software, algorithm | g:Profiler | *Raudvere et al., 2019* | | |
| Software, algorithm | REVIGO | http://revigo.irb.hr/ | RRID:SCR_005825 | |
| Software, algorithm | Python | https://www.python.org/ | RRID:SCR_008394 | Version 2.7.14 and 3.6.12 |
| Software, algorithm | R | https://www.r-project.org/ | RRID:SCR_001905 | Version 4.0.2 |

## Generating *Drosophila* α-arrestin-GFP fusion DNA constructs

To create *Drosophila* ARRDC entry clones, we gathered cDNA sequences of 12 *Drosophila* α-arrestins: CG2993 (#2276, Drosophila Genomics Resource Center [DGRC], Bloomington, IN), CG18744 (#1388606, DGRC), CG18745 (#12871, DGRC), CG18746 (#9217, DGRC), CG18747 (#1635366, DGRC), CG18748 (#1387253, DGRC), CG2641 (#1649402, DGRC), CG10086 (#8816, DGRC), CG14696 (#1644977, DGRC), CG1105 (#4234, DGRC), Vdup1 (#1649326, DGRC), and Leash (*Kwon et al., 2013*). We then subcloned each cDNA sequence of *Drosophila* α-arrestins into pCR8 entry clone vector using pCR8/GW/TOPO TA cloning kit (#K250020, Thermo Fisher Scientific, Waltham, MA) by following the manufacturer's protocol. To generate plasmids with suitable system for protein expression in *Drosophila* cell culture, we then subcloned these α-arrestins-containing-pCR8 plasmids into pMK33-Gateway-GFP destination vector (*Kwon et al., 2013*; *Kyriakakis et al., 2008*) using Gateway LR Clonase II enzyme mix (#11791020, Thermo Fisher Scientific), where coding sequences of α-arrestins are inserted before GFP sequence. Final constructs were validated using GENEWIZ Sanger Sequencing.

## Establishing *Drosophila* α-arrestin-GFP stably expressing cell lines

S2R+ cells (*Schneider, 1972*; stock number: 150; DGRC) were maintained in Schneider's *Drosophila* Medium (#21720024, Thermo Fisher Scientific) supplemented with 10% heat-inactivated fetal bovine serum (FBS, #16140071, Thermo Fisher Scientific) and 1% penicillin–streptomycin (#15070063, Thermo Fisher Scientific) at 24°C. To establish α-arrestin-GFP stably expressing *Drosophila* cell lines, $0.4 \times 10^6$ S2R+ cells were seeded in six-well plates and transfected with 1 µg of each pMK33-ARRDC-GFP construct using Effectene transfection reagent (#301425, QIAGEN, Venlo, the Netherlands). pMK33 plasmid is a copper-induced protein expression vector, which carries Hygromycin B-antibiotic-resistant gene. Therefore, we selected α-arrestin-GFP stable cell lines by maintaining cells in Schneider's *Drosophila* Medium supplemented with 200 µM Hygromycin B (#40-005, Thermo Fisher Scientific). The stable cells were transferred into T25 cm² flasks to repopulate. To induce the expression of α-arrestin-GFP fusion proteins, we exposed the stable cells to 500 µM $CuSO_4$ (#C8027, Sigma-Aldrich, Burlington, MA) to the media. We confirmed the GFP-tagged α-arrestin protein expressions using fluorescence microscopy.

## Synthesizing human α-arrestin coding sequence

Due to the lack of commercially available stock, we utilized GENEWIZ (South Plainfield, NJ) gene synthesis service to synthesize human ARRDC5 coding sequence (NM_001080523).

## Generating mammalian GFP-α-arrestin fusion DNA constructs

To create human α-arrestin entry clones, we subcloned ARRDC3 (#38317, Addgene, Watertown, MA) and ARRDC5 (GENEWIZ) into pCR8 entry clone vector using pCR8/GW/TOPO TA cloning kit (#K250020, Thermo Fisher Scientific) by following the manufacturer's protocol. ARRDC1 (BC032346, GeneBank), ARRDC2 (BC022516, GeneBank), ARRDC4 (BC070100, GeneBank), and TXNIP (BC093702, GeneBank) were cloned into pCR8. To generate plasmids with suitable system for protein expression in mammalian cell culture, we then subcloned these α-arrestin s-containing-pCR8 plasmids into pHAGE-GFP-Gateway destination vector (gift from Dr. Chanhee Kang at Seoul National University) using Gateway LR Clonase II enzyme mix (#11791020, Thermo Fisher Scientific), where coding sequences of α-arrestin are inserted after GFP sequence. Final constructs were validated using GENEWIZ Sanger Sequencing.

## Establishing mammalian GFP-α-arrestin stably expressing cell lines

We produced GFP-α-arrestins lentiviral particles by seeding $5 \times 10^6$ HEK293T (CRL-3216; American Type Culture Collection [ATCC], Manassas, VA) cells in 10 cm² dish with Dulbecco's Modified Eagle Medium (DMEM, #11965118, Thermo Fisher Scientific) supplemented with 25 mM HEPES, 10% heat-inactivated FBS (#16140071, Thermo Fisher Scientific), and 1% penicillin–streptomycin (#15070063, Thermo Fisher Scientific) at 37°C in humidified air with 5% $CO_2$. Approximately after 16–24 hr, at 90% cell confluency, we changed the cell media to Opti-MEM medium (#31985070, Thermo Fisher Scientific) and transfected the cells with 10 µg pHAGE-GFP-α-arrestin construct, 10 µg lentivirus packaging plasmid (pCMV-dR8.91), and 10 µg virus envelope plasmid (VSV-G) using PEIPro DNA

transfection reagent (#115010, VWR, Radnor, PA). GFP-α-arrestins lentiviral particles were harvested 40 hr post transfections. To establish GFP-α-arrestins stably expressing mammalian cell lines, HEK293 (CRL-1573; ATCC) cells were seeded in 10 cm$^2$ dish with DMEM (#11965118, Thermo Fisher Scientific) supplemented with 25 mM HEPES, 10% heat-inactivated FBS (#16140071, Thermo Fisher Scientific), and 1% penicillin–streptomycin (#15070063, Thermo Fisher Scientific) at 37°C in humidified air with 5% $CO_2$. At 90% cell confluency, cells were infected with pHAGE-GFP-ARRDC lentivirus particle, and stable cells were selected by maintaining cells in media supplemented with 1.5 µg/mL puromycin (#BP2956100, Thermo Fisher Scientific). We confirmed the GFP-tagged α-arrestin protein expressions using fluorescence microscopy.

## Immunofluorescence imaging of human α-arrestins

Stably α-arrestin-GFP expressing HEK293 cells were cultured in a 12-well plate with pre-sterilized round glass coverslips in each well. Cells on coverslip were fixed in 4% paraformaldehyde (PFA; RT15710, Electron Microscopy Sciences, Hatfiled, PA) diluted in PBS for 30 min and then washed three times with PBST (PBS supplemented with 0.2% Triton X-100) with 5 min intervals. To label the nucleus, samples were stained with DAPI (1:5000; D9542, Sigma-Aldrich) in PBST supplemented with 1% BSA (A7906, Sigma-Aldrich) for 1 hr at room temperature. Stained cells samples were washed three times with PBST and preserved in Vectashield (H-1000, Vector Laboratories, Burlingame, CA). Fluorescence images were acquired using an Olympus FV1200 confocal microscope with 40× oil objective lens and 2× zoom factor. NIH ImageJ software was used for further adjustment and assembly of the acquired images.

## Affinity purification of *Drosophila* and human GFP-tagged α-arrestin complexes

We seeded each of the *Drosophila* α-arrestin-GFP stable cells in six T-75 cm$^2$ flasks (2.1 × 10$^6$ cells per flask), and α-arrestin-GFP expression was induced for 48 hr with 500 µM CuSO$_4$. Meanwhile, we seeded each of the human GFP-α-arrestin stable cells in eight T-75 cm$^2$ flasks and grown for 48 hr before collection. The cells were harvested by spinning down cells at 1000 × *g* for 5 min and washed once with cold PBS. We lysed the cells by resuspending cells in lysis buffer (10 mM Tris-HCl pH 7.5, 150 mM NaCl, 0.5 mM EDTA, 1.5 mM MgCl$_2$, 5% glycerol, 0.5% NP-40, 25 mM NaF, 1 mM DTT, and 1× HALT protease and phosphatase inhibitor [#PI78442, Thermo Fisher Scientific]) and incubating them for 40 min. The lysate was separated from the insoluble fraction by centrifugation at 20,000 × *g* for 15 min at 4°C. To capture the α-arrestins and their native interactors, each α-arrestin-containing lysate was incubated with GFP-nanobody-conjugated to Dynabeads M-270 Epoxy magnetic beads (#14301, Thermo Fisher Scientific). The supernatant was separated from the beads using a magnetic rack, and the beads were washed five times with lysis buffer. The protein complexes were eluted from the beads by adding 200 mM glycine pH 2.5, and the pH was neutralized with Tris base pH 10.4. Purified α-arrestin proteins were confirmed by running a fraction of the eluted proteins on SDS-PAGE/Coomassie gel.

## Protein sample preparation for mass spectrometry

To digest protein samples into peptides for mass spectrometry analysis, we precipitated the eluted proteins by adding trichloroacetic acid (#T0699, Sigma-Aldrich) to 20% final concentration, followed by spinning down samples at maximum speed for 30 min at 4°C. The precipitates were washed with 10% trichloroacetic acid solution and three additional times with acetone (#A929, Thermo Fisher Scientific), and left to dry at room temperature. Protein precipitations were digested with Trypsin (#V5113; Promega, Madison, WI) diluted in Digestion buffer (100 mM ammonium bicarbonate and 10% acetonitrile) in a 1:40 ratio. Resulting peptides were purified using ZipTip Pipet tips (#ZTC18M096, Thermo Fisher Scientific).

## LC/MS-MS analysis

We used cells stably expressing GFP and wild-type HEK293 or S2R+ cells alone as control baits. AP/MS experiments for all *Drosophila* and human α-arrestin baits were performed in two biological replicates, with the exception of human ARRDC3 baits (two technical replicates). Samples were resuspended in Mass Spectrometry buffer (5% formic acid and 5% acetonitrile) and analyzed on a Liquid

Chromatography Orbitrap Fusion Lumos Tribrid Mass Spectrometer (#IQLAAEGAAPFADBMBHQ, Thermo Fisher Scientific) equipped with a nano-Acquity UPLC system and an in-house-developed nano spray ionization source. Peptides were separated using a linear gradient, from 5 to 30% solvent B (LC-MS grade 0.1% formic acid [#A117, Thermo Fisher Scientific] and acetonitrile) in a 130 min period at a flow rate of 300 nL/min. The column temperature was maintained at a constant 5°C during all experiments. Peptides were detected using a data-dependent method. Survey scans of peptide precursors were performed in the Orbitrap mass analyzer from 380 to 1500 m/z at 120 K resolution (at 200 m/z), with a $5 \times 10^5$ ion count target and a maximum injection time of 50 ms. The instrument was set to run in top speed mode with 3 s cycles for the survey and the MS/MS scans.

## Database searching and analysis of mass spectrometry data

MS/MS spectra were queried using the Comet search engine (*Eng et al., 2013*) to search for corresponding proteins in FlyBase (*Gramates et al., 2017*) and UniProt (*The UniProt Consortium, 2017*). Common contaminant protein sequences from the Common Repository of Adventitious Proteins (cRAP) Database (ftp://ftp.thegpm.org/fasta/cRAP) were used to filter contaminating sequences. Searching was done with the following parameters: tryptic digest, internal decoy peptides, the number of missed cleavages = 2, precursor tolerance allowing for isotope offsets = 20 ppm, a 1.00 fragment bin tolerance, static modification of 57.02 on cysteine, and variable modification of 16.00 on methionine. The acetylation, phosphorylation, and ubiquitination searches add variable modifications of 42.01 on lysine, 79.97 on serine/threonine/tyrosine, and 114.04 on lysine, respectively. The search results were then processed using the Trans-Proteomic Pipeline suite of tools version 4.8.0 (*Keller et al., 2005*), where the PeptideProphet tool (*Keller et al., 2002*) was applied to calculate the probability that each search result is correct and the ProteinProphet tool (*Nesvizhskii et al., 2003*) was applied to infer protein identifications and their probabilities.

## TXNIP knockdown in HeLa cells

HeLa cells (CCL-2; ATCC) were cultured in complete DMEM supplemented with 10% FBS and 1% penicillin–streptomycin. Cells were cultured in an incubator at 37°C in humidified air containing 5% $CO_2$. For siRNA-induced knockdown of TXNIP in HeLa cells, the following siRNA duplex was synthesized (Bioneer, Daejeon, South Korea): sense: 5′-GUCAGUCACUCUCAGCCAUdTdT-3′, anti-sense: 5′-AUGGCUGAGAGUGACUGACdTdT-3′. Random sequence siRNAs (AccuTarget Negative control siRNA; Bioneer), which are non-targeting siRNAs that have low sequence homology with all humans, mouse, and rat genes, were used as negative controls (siCon). Then, 100 nM of each siRNA was transfected into 105 HeLa cells using Lipofectamine RNAiMAX (#13778075, Invitrogen, Carlsbad, CA) according to the manufacturer's instructions. Transfected cells were harvested after 48 hr for RNA-seq and ATAC-seq (two biological replicates for each sequencing data).

## RNA sequencing

For RNA-seq, total RNA was extracted using TRIzol (#15596018, Invitrogen) according to the manufacturer's protocol. Total RNA concentration was calculated using Quant-IT RiboGreen (#R11490, Invitrogen). To assess the integrity of the total RNA, samples are run on the TapeStation RNA screentape (#5067-5576, Agilent Technologies, Santa Clara, CA). Only high-quality RNA preparations, with RNA integrity number greater than 7.0, were used for RNA library construction. A library was independently prepared with 1 ug of total RNA for each sample using the Illumina TruSeq Stranded mRNA Sample Prep Kit (#RS-122-2101, Illumina, Inc, San Diego, CA). The first step in the workflow involves purifying the poly-A-containing mRNA molecules using poly-T-attached magnetic beads. Following purification, the mRNA is fragmented into small pieces using divalent cations under elevated temperature. The cleaved RNA fragments are copied into first-strand cDNA using SuperScript II reverse transcriptase (#18064014, Invitrogen) and random primers. This is followed by second-strand cDNA synthesis using DNA Polymerase I, RNase H, and dUTP. These cDNA fragments then go through an end repair process, the addition of a single 'A' base, and then ligation of the adapters. The products are then purified and enriched with PCR to create the final cDNA library. The libraries were quantified using KAPA Library Quantification kits (#KK4854, KAPA BIOSYSTEMS, Wilmington, MA) for Illumina Sequencing platforms according to the qPCR Quantification Protocol Guide and qualified using the TapeStation D1000 ScreenTape (#5067-5582, Agilent Technologies). Indexed libraries were then submitted to an

Illumina NovaSeq 6000 (Illumina, Inc) as the paired-end (2×100 bp) sequencing. Both library preparation and sequencing were performed using Macrogen (Macrogen, Inc, Seoul, South Korea).

## ATAC sequencing

A total of 100,000 cells were prepared using LUNA-FL Automated Fluorescence Cell Counter (#L20001, Logos Biosystems, Gyeonggi-do, South Korea). Cells were lysed using cold lysis buffer, which consist of nuclease-free water (#10977023, Invitrogen), IGEPAL CA-630 (#I8896, Sigma-Aldrich), 1 M Trizma HCl (pH 7.4) (#T2194, Sigma-Aldrich), 5 M NaCl (#59222C, Sigma-Aldrich), and 1 M $MgCl_2$ (#M1028, Sigma-Aldrich). The nuclei concentration was determined using Countess II Automated Cell Counter (#AMQAX1000, Thermo Fisher Scientific), and nuclei morphology was examined using microscopy. Immediately after lysis, resuspended nuclei (50,000 cells) were put in transposition reaction mix 50 µL, which consist of TED1 2.5 µL and TD 17.5 µL (#20034197, Illumina, Inc), nuclease-free water 15 µL, and the nuclei resuspension (50,000 nuclei, 15 µL). The transposition reaction was incubated for 30 min at 37°C. Immediately following transposition, the products were purified using a MinElute PCR purification Kit (#28004, QIAGEN). Next, transposed DNA fragments were amplified using Nextera DNA Flex kit (#20018704, Illumina, Inc). To reduce GC and size bias in PCR, the appropriate number of cycles was determined as follows: qPCR side reaction was run, the additional number of cycles needed was calculated, liner Rn versus cycle was plotted, and the cycle number that corresponds to 1/4 of maximum fluorescent intensity was determined. The remaining PCR reaction was run to the cycle number determined. Amplified library was purified and then quantified using KAPA library quantification kit (#07960255001, Roche, Basel, Switzerland) and Bioanalyzer (Agilent Technologies). The resulting libraries were sequenced using HiSeq X Ten (Illumina, Inc). Both library preparation and sequencing were performed using Macrogen (Macrogen, Inc).

## Immunoblotting and co-immunoprecipitation assays

Cells were lysed in radioimmunoprecipitation assay (RIPA) buffer supplemented with protease inhibitor. For immunoblotting, the cell lysates were separated by 4–20% SDS-polyacrylamide gel electrophoresis (PAGE) and transferred to nitrocellulose membranes. After blocking membranes with 5% skim milk in Tris buffered saline containing 0.1% Tween-20 (TBS-T) for 1–2 hr at room temperature, the nitrocellulose membranes were incubated with appropriate primary antibodies overnight at 4°C and subsequently reacted with horseradish peroxidase (HRP)-conjugated secondary antibodies for 1 hr at room temperature. Bands were visualized using an enhanced chemiluminescence (ECL) detection system, West-Q Pico ECL Solution (W3652-02, GenDEPOT, Katy, TX) or Fusion FX Spectra (Vilber, Marne-la-Vallée, France). For quantification of immunoblot results, the densities of target protein bands were analyzed with ImageJ.

For immunoprecipitation, the cell lysates (2 mg) were incubated with appropriate antibodies (1 µg) overnight at 4°C and precipitated with TrueBlot Anti-Rabbit Ig IP agarose beads (00-8844-25; Rockland, Philadelphia, PA) for 2 hr at 4°C. The immunocomplexes were washed with chilled PBS three times and heated with 3× sample loading buffer containing ß-mercaptoethanol. The samples were separated by 6–8% SDS-PAGE, and immunoblot was performed as described above.

The following antibodies were used for immunoblotting and co-immunoprecipitation assays: anti-TXNIP (#14715), anti-HDAC2 (#57156), anti-alpha Tubulin (#3873), anti-phospho-HDAC2 (#69238), anti-MTA1 (#5647), anti-MBD3 (#14540), anti-ATP6V1B2 (14617S), HRP-linked anti-rabbit IgG (7074S), and HRP-linked anti-mouse IgG (7076S) were obtained from Cell Signaling Technology (Beverly, MA); anti-H3ac (39139) was obtained from Active Motif (Carlsbad, CA); anti-ß-actin (GTX629630) was obtained from GeneTex; normal anti-rabbit IgG (sc-2027) and anti-GAPDH (sc-365062) were obtained from Santa Cruz Biotechnology (Dallas, TX); and TrueBlot anti-rabbit IgG HRP (18-8816-31) was obtained from Rockland (Philadelphia, PA).

## Quantitative reverse-transcription polymerase chain reaction (PCR)

Total RNA was isolated using TRIzol reagent (#15596018, Invitrogen) and subjected to reverse transcription PCR (RT-PCR) with ReverTra Ace qPCR RT kit (#FSQ-101, Toyobo, Osaka, Japan) or GoScript RT-PCR system (#A5001, Promega) according to the manufacturer's instructions. The mRNA expression levels of target genes were quantified using the CFX Opus 96 (Bio-Rad, Hercules, CA) or Applied Biosystems QuantStudio 1 (Applied Biosystems, Foster City, CA) real-time PCR. AccuPower 2X

GreenStar qPCR Master Mix (#K6251, Bioneer) or SYBR Green Realtime PCR Master Mix (#QPK-201, Toyobo) were applied according to the manufacturer's protocols. The data was normalized by GAPDH or alpha-tubulin mRNA levels and calculated using the ΔΔCt method (*Hellemans et al., 2007*). The primers used for qRT-PCR analysis are summarized in the Key resources table.

## Immunofluorescence of HDAC2 and TXNIP

HeLa cells were cultured in six-well plates with cover slips in each well ($1.5 \times 10^4$ cells/well). After cells were incubated overnight in Opti-MEM, TXNIP knockdown was induced by transfection of siRNA at a concentration of 100 nM. Following 48 hr of transfection, the cells were washed twice with PBS and then fixed with 100% ice-cold methanol for 10 min at –20°C. After rinsing three with PBSTw (PBS containing 0.1% Tween 20), the cells were blocked with 3% BSA in PBS and incubated for 45 min at room temperature. Next, cells were incubated with the primary antibody for 150 min followed by the secondary antibody for 60 min in the dark. For co-staining with a second primary antibody, the blocking step followed by the primary and secondary antibody incubation steps were repeated. All of the antibodies were diluted in antibody dilution buffer (1% BSA in PBS). Information of the antibodies is listed in the Key resources table. The cover slips were rinsed three times with PBSTw and then mounted with VECTASHIELD Antifade Mounting Medium containing DAPI (Vector Laboratories) according to the manufacturer's instructions. The fluorescence was visualized with a Nikon C2 Si-plus confocal microscope. Fluorescence images were observed under a ZEISS confocal microscope (LSM5; Carl Zeiss, Jena, Germany), and the integrated densities of fluorescence were analyzed using ImageJ program.

## Nuclear–cytoplasmic fractionation

Prior to transfection, HeLa cells were seeded in 100 mm cell culture dishes containing Opti-MEM medium and incubated overnight (reaching a confluency of approximately 30–40%). The cells were then transfected with siTXNIP. Cells were harvested after 48 hr of transfection and fractionated according to the manufacturer's instructions using NE-PER Nuclear and Cytoplasmic Extraction Reagents (#78833, Thermo Fisher Scientific). Protease inhibitor cocktail (P8340; Sigma-Aldrich) was added as a supplement to the lysis buffer, and the protein concentration was measured using a Pierce BCA Protein Assay Kit (#23225, Thermo Fisher Scientific).

## ChIP assay

Cells were crosslinked with 1% formaldehyde at 37°C or room temperature for 15 min and the reaction was stopped by the addition of 0.125 M glycine. ChIP was then performed using a ChIP-IT High Sensitivity kit (#53040, Active Motif) according to the manufacturer's instructions. Enrichment of the ChIP signal was detected by quantitative real-time PCR (qPCR). The data of each biological replicate were normalized with negative control IgG signals, and enrichment values were calculated using the ΔΔCt method (*Hellemans et al., 2007*). The following antibodies were used: TXNIP (14715, Cell Signaling Technology, Beverly, MA), HDAC2 (57156, Cell Signaling Technology), H3ac antibody (39139; Active Motif), and normal rabbit IgG antibodies. The primers used for ChIP-qPCR are summarized in the Key resources table.

## Osteoclast differentiation and collection of lentiviruses for ARRDC5 expression

BMMs were cultured as previously described (*Kim et al., 2019b*). Briefly, bone marrow was obtained from mouse femurs and tibias at 8 wk of age (C5BL/6; KOATECH, Gyeonggi-do, South Korea; *Supplementary file 12*), and BMMs were isolated from the bone marrow using Histopaque (1077; Sigma-Aldrich). BMMs were seeded at a density of $1.2 \times 10^5$ cells/well into 24-well culture plates and incubated in α-MEM (SH30265.01; Hyclone, Rockford, IL) containing 20 ng/mL macrophage colony-stimulating factor (M-CSF; 300-25; PeproTech, Cranbury, NJ). To induce osteoclast differentiation, BMMs were treated for 24 hr with lentiviral-containing medium that also contained M-CSF, after which the medium was changed to α-MEM containing 20 ng/mL M-CSF and 20 ng/mL RANKL (462-TEC; R&D Systems, Minneapolis, MN). The differentiation medium was changed every 24 hr during the 5-day differentiation period.

To obtain the media containing lentivirus, HEK293 cells were cultured in DMEM containing 4.5 g/L glucose (SH30243.01; Hyclone) supplemented with 10% FBS (SH30084.03; Hyclone) and 1% penicillin–streptomycin. After seeding cells at a density of $1 \times 10^5$ cells/well into six-well culture plates, the cells were incubated with lentivirus co-transfected media for 16 hr. Lentivirus co-transfected media was prepared according to the manufacturer's instructions using the CRISPR & MISSION Lentiviral Packaging Mix (SHP002; Sigma-Aldrich) and the lentiviral transfer vector. The lentiviral vector pHAGE-GFP-ARRDC5 was used to generate GFP-ARRDC5-overexpressing osteoclasts, and pHAGE-GFP-GFP served as the control vector. The pLKO.1-puro-CMV-tGFP vector (SHC003; Sigma-Aldrich) was employed to generate ARRDC5 knockdown, which contains the target sequence 5'-CCACACCT TTGAACTTCCATTT-3', and this vector was also used to generate its non-target control. After the incubation, the medium was replaced with fresh α-MEM medium supplemented with 10% FBS and 1% penicillin–streptomycin. The medium was collected twice (after 24 and 48 hr), designated as lentiviral-containing medium, and stored in a deep freezer until used to infect BMMs.

## TRAP staining and bone resorption pit assay

Osteoclast differentiation and activity were determined by TRAP staining and a bone resorption pit assay, respectively. TRAP staining was performed using a TRAP staining kit (PMC-AK04F-COS; Cosmo Bio Co., Ltd, Tokyo, Japan) following the manufacturer's instructions. TRAP-positive multinucleated cells with more than three nuclei were counted under a microscope using ImageJ software (NIH, Bethesda, MD). The bone resorption pit assay was performed using dentin discs (IDS AE-8050; Immunodiagnostic Systems, Tyne & Wear, UK). Cells were differentiated to osteoclasts on the discs over a 4-day period, after which the discs were stained with 1% toluidine blue solution and the resorption pit area was quantified using ImageJ software.

## Immunofluorescence staining of the V-type ATPase and visualization with GFP-ARRDC5

To inhibit V-type ATPase transport to the membrane (*Matsumoto and Nakanishi-Matsui, 2019*), osteoclasts on the fifth day of differentiation were incubated with 100 nM bafilomycin A1 (19-148; Sigma-Aldrich) for 3 hr. Then, immunofluorescence staining was performed to visualize the localization of the V-type ATPase in bafilomycin A1-treated and untreated cells. The cells were fixed using a 4% PFA solution (PC2031-100; Biosesang, Gyeonggi-do, Korea) and permeabilized using 0.05% Triton X-100 at room temperature for 5 min. The cells were incubated with anti-V-type ATPase antibody (SAB1402125-100UG; Sigma-Aldrich) at room temperature for 1 hr, and then stained with the Alexa Fluor 594-conjugated anti-mouse antibody (A-21044; Invitrogen) at room temperature for 30 min. Finally, cells were mounted using Antifade Mountant with DAPI (P36962; Invitrogen). Fluorescence images were observed under a ZEISS confocal microscope (LSM5; Carl Zeiss).

## Cell line authentication and mycoplasma contamination test

We newly purchased HEK293, HEK293T, and S2R+ (DGRC Stock 150; https://dgrc.bio.indiana.edu//stock/150; RRID:CVCL_Z831) cell lines from either ATCC or the DGRC for the study. Detailed information of the cell lines is summarized in the Key resources table. After establishing frozen stocks, these cell lines were immediately used. Note that the short tandem repeat (STR) profiles for HEK293 and HEK293T cell lines are available from ATCC, while the profile for the S2R+ cell line is accessible through the DGRC (*Schneider, 1972*). HeLa cell, a gift from Dr. Jungwook Hwang at Hanyang University and originally sourced from ATCC, was re-authenticated using STR profiling by the biotechnology company Macrogen. The STR analysis report for HeLa cell is included in *Supplementary file 12*. We routinely performed mycoplasma contamination tests with PCR-based mycoplasma kits, such as Mycoplasma PCR Detection Kit (G238; abm, Vancouver, Canada) or e-Myco plus Mycoplasma PCR Detecting Kit (25237; iNtRON Biotechnology, Seoul, South Korea), and the test results are included in *Supplementary file 12*.

## Functional annotations and multiple sequence alignment of α-arrestin sequences

The sequences of 12 *Drosophila* and 6 human α-arrestins were retrieved from the UniProt database (*The UniProt Consortium, 2017*). Domains and motifs including the PPxY motif were annotated

based on sequences from Pfam version 31.0 (*El-Gebali et al., 2019*) and the ELM database (*Dinkel et al., 2016*). The sequences were subjected to the multiple-sequence alignment tool T-COFFEE (*Notredame et al., 2000*) using default parameters. The output of T-COFFEE was applied to RAxML (version 8.2.11; *Stamatakis, 2014*) to generate a consensus phylogenetic tree with 1000 rapid boot-strapping using '-m PROTGAMMAWAGF' as the parameter.

## Identification of high-confidence bait–prey PPIs

### SAINTexpress analysis

To identify high-confidence bait–prey PPIs, spectral counts of AP/MS data from S2R+ and HEK293 cells were subjected to the SAINTexpress algorithm (version 3.6.1; *Teo et al., 2014*), which calculates the probability of authenticity for each bait–prey PPI. The program outputs the SAINTexpress scores and the Bayesian false discovery rates (BFDR) based on the spectral count distribution of true and false PPI sets. Before calculating the scores, bait-to-bait self-interactions were removed manually. SAINTexpress was run with the '-R 2' parameter, which specifies the number of replicates, and the '-L 3' parameter, which specifies the number of representative negative control experiments to be considered.

### PPI validation datasets

To evaluate the performance of the PPI prediction based on the SAINTexpress score, validation data-sets including positive and negative PPIs were precompiled as described in previous studies (*Kwon et al., 2013*; *Vinayagam et al., 2016*). Briefly, the positive PPIs were initially collected by searching for known PPIs involving α-arrestins from STRING version 10.5 (https://string-db.org/; *Szklarczyk et al., 2015*), GeneMANIA version 3.4.1 (*Warde-Farley et al., 2010*), Bioplex (*Huttlin et al., 2015*), and DpiM (*Guruharsha et al., 2011*). For human, additional positive PPIs were curated from the literature (*Colland et al., 2004*; *Dotimas et al., 2016*; *Nabhan et al., 2012*; *Nishinaka et al., 2004*; *Puca and Brou, 2014*; *Wu et al., 2013*). After these steps, 30 PPIs (21 preys) for human and 46 PPIs (17 preys) for *Drosophila* were considered as positive PPIs (*Supplementary file 2A and C*). Proteins manually curated from the Contaminant Repository for Affinity Purification (CRAPome) (*Mellacheruvu et al., 2013*) were compared to those detected in our negative controls, and only those that were detected in both were considered as were negative PPIs (*Supplementary file 2B and D*). As a result of these steps, 1372 PPIs (268 preys) for human and 1246 PPIs (122 preys) for *Drosophila* were compiled as negative PPIs.

### Construction of high-confidence PPI networks

The performance of SAINTexpress was evaluated using the positive and negative PPIs. Because there is an imbalance between positive and negative PPIs, 1000 random cohorts of negative PPIs number-matched with that of positive PPIs were generated. The average true-positive and false-positive rates were plotted as ROC curves over different SAINTexpress scores as a cutoff, and AUC values were calculated using the ROCR R package (version 1.0-11, https://cran.r-project.org/web/packages/ROCR/index.html). Based on these results, we chose an optimal cutoff for high-confidence PPIs with a BFDR of 0.01, where the false-positive rates were less than 3% (~1.8% for human and ~2.7% for *Drosophila*) in both species, and the true-positive rates were substantially higher (~66.7% for human and ~45.7% for *Drosophila*). The cutoffs correspond to SAINTexpress scores of 0.85 and 0.88 for human and *Drosophila*, respectively.

## Checking the reproducibility of spectral counts among replicates

If multiple proteins isoforms were detected, they were collapsed into a single gene. To avoid the divide-by-zero error, spectral counts of '0' were converted to a minimum non-zero value, '0.01'. To examine the integrity and quality of spectral counts from the AP/MS, the average correlation coefficients (Pearson) of spectral counts from α-arrestins were calculated and plotted. At each cutoff of spectra counts from 1 to 15, only the PPIs with spectral counts that were the same or higher than the cutoff for all replicates were kept and used to calculate correlation coefficients between replicates. The resulting coefficients from the α-arrestin interactomes were then averaged and plotted. At the cutoff of six spectral counts, saturation of average correlation coefficients was observed and chosen

as an optimal cutoff to filter the PPIs. PCA of the filtered PPIs was conducted based on spectral counts (with a pseudo count 1 added) transformed into a $\log_2$ using the factoextra R package (version 1.0.7).

### Hierarchical clustering of high-confidence PPIs

Hierarchical clustering based on $\log_2$ spectral counts (pseudo count 1 added) of high-confidence PPIs was conducted using the Pearson correlation as the clustering distance and Ward's method as the clustering method. Heatmaps were visualized using the ComplexHeatmap R package (version 2.6.2; *Gu et al., 2016*). Six clusters were identified for each species based on the results of hierarchical clustering; the PANTHER protein class overrepresentation test was performed for the proteins in each cluster (*Thomas et al., 2003*). FDRs (Fisher's exact test) of indicated protein classes were ≤0.05 for all classes except for 'GTPase-activating protein' in human (FDR < 0.133) and 'GEFs' in *Drosophila* (FDR < 0.109), respectively. Interacting prey proteins from the positive PPIs were selectively labeled.

### Domain and motif analysis of bait and prey proteins

For human and *Drosophila*, respectively, 53 and 65 short-linear motifs in α-arrestins were annotated using the ELM database (*Dinkel et al., 2016*), and 423 and 546 protein domains in prey proteins were annotated using the UniProt database (*The UniProt Consortium, 2017*; *Supplementary file 4*). To test for enrichment of protein domains, we implemented the Expression Analysis Systematic Explorer (EASE) score (*Hosack et al., 2003*), which is calculated by subtracting one gene within the query domain and conducting a one-sided Fisher's exact test. Protein domains enriched in the interactomes of each α-arrestin (Benjamini–Hochberg FDR ≤ 0.05) were plotted using the ComplexHeatmap R package (version 2.6.2). Next, to see how reliable our filtered PPIs were, we utilized information about known affinities between domains and short-linear motifs from the ELM database (*Dinkel et al., 2016*). Because the arrestin_N (Pfam ID: PF00339) and arrestin_C (Pfam Id: PF02752) domains in α-arrestins do not have known interactions with any of the short-linear motifs in the ELM database (*Dinkel et al., 2016*), only the interactions between the short-linear motifs in α-arrestins and protein domains in the interactome (prey proteins) were considered in this analysis. We found that 59 out of the 390 human PPIs and 64 out of the 740 *Drosophila* PPIs were supported by such known affinities (*Supplementary file 4*). One-sided Fisher's exact test was used to test the significance of the enrichment of the supported PPIs in the filtered PPI sets versus those in the unfiltered PPI sets (*Figure 1D*).

### Subcellular localizations of bait and prey proteins

To search for annotated subcellular localizations of the proteins in the α-arrestin interactomes, we first obtained annotation files of cellular components (GO: CC) for human and *Drosophila* from the Gene Ontology Consortium (*Ashburner et al., 2000*). From the annotations, we only utilized GO terms for 11 subcellular localizations (name of subcellular localization – GO term ID: Cytosol – GO:0005829; Plasma membrane – GO:0005886; Nucleus – GO:0005634; Mitochondrion – GO:0005739; Endoplasmic reticulum – GO:0005783; Golgi apparatus – GO:0005794; Cytoskeleton – GO:0005856; Peroxisome – GO:0005777; Lysosome – GO:0005764; Endosome – GO:0005768; Extracellular space – GO:0005615). If a protein was annotated to be localized in multiple locations, a weighted value (1/ the number of multiple localizations) was assigned to each location. Finally, the relative frequencies of the subcellular localizations associated with the interacting proteins in the filtered PPIs were plotted for each α-arrestin (*Figure 1—figure supplement 3B*).

### Identification of protein complexes associated with α-arrestins

To examine protein complexes significantly enriched in the α-arrestin interactomes, we collected known protein complexes from two databases: COMPLEAT (*Vinayagam et al., 2013*), which is a comprehensive resource of protein complexes built from information in the literature and predicted by orthologous relationships of proteins across species (human, *Drosophila*, and yeast), and the DAVID GO analysis of cellular components (*Huang et al., 2009a*; Benjamini–Hochberg FDR ≤ 0.05; *Supplementary file 7B and D*), from which bulk cellular compartments such as the nucleus, cytosol, and so on were excluded. From the COMPLEAT database, we evaluated the association of the resulting protein complexes with each α-arrestin by the complex association score, which is the IQM of SAINTexpress scores (*Equation 1*)

$$Complex\,association\,score\,(IQM) = \frac{\sum_{i=Q1}^{Q3} SAINTexpress\,score_i}{(Q3 - Q1) + 1} \tag{1}$$

where the first quartile is $Q1 = \frac{N}{4} + 1$, the third quartile is $Q3 = \frac{3N}{4}$, and $N$ is the total number of preys in the complex. The significance of the complex association score was estimated by comparing the score to the null distribution of the scores calculated from 1000 random complexes of input proteins. The significance was tested using the online COMPLEAT tool, and protein complexes with p<0.05 were selected for further analysis (*Supplementary file 7A and C*). Next, we iteratively combined (clustered) the pairs of protein complexes from any two databases (COMPLEAT and GO analysis of cellular components) that showed the highest overlap coefficients, $Overlap\,(X, Y)$ (*Equation 2*; *Vijaymeena and Kavitha, 2016*), until there was no pair of complexes whose coefficients were higher than 0.5.

$$Overlap\,(X, Y) = \frac{|X \cap Y|}{(|X|, |Y|)} \tag{2}$$

From the clustered set of complexes, we manually removed those with fewer than three subunits or two PPIs. Subunits in the complexes that have no connection among themselves were also removed. Lastly, the significance of associations of the resulting complexes with each α-arrestin was tested in the same manner as done in COMPLEAT using complex association score. The resulting p-values were corrected using the Benjamini–Hochberg procedure, and only interactions with statistical significance (FDR < 0.2) were visualized with Cytoscape v3.5.1 (*Shannon et al., 2003*; *Figures 2 and 3*, *Figure 2— figure supplement 1*, *Figure 3—figure supplement 1*).

## Orthologous networks of α-arrestin interactomes

DIOPT (version 7.1) was used to search for orthologs of all prey proteins and only those with a DIOPT score ≥ 2 were selected for the identification of orthologous PPIs between *Drosophila* and human. Next, the orthologs were tested for the enrichment of GO biological process and molecular functions and Kyoto Encyclopedia of Genes and Genomes pathway using DAVID (*Huang et al., 2009a*). In addition, manual curation of individual genes was performed through the UniProt database (*The UniProt Consortium, 2017*). The orthologs were manually grouped into functional modules based on the results, and α-arrestins were modularized into seven groups based on hierarchical clustering of $\log_2$-transformed mean spectral counts using the correlation distance and the Ward linkage method. The heatmap was plotted using the pheatmap R package (version 1.0.12).

## Processing of RNA-seq data

For quality checks and read trimming, RNA-seq data were processed by FastQC (version 0.11.8; *Andrews, 2010*) and sickle (version 1.33; *Joshi and Fass, 2011*) with default parameters. After the trimming, the reads were aligned to human transcriptomes (GENCODE version 29, GRCH38/hg38; *Frankish et al., 2019*) using STAR (version 2.5.3a_modified; *Dobin et al., 2013*) with default parameters and read counts were determined using RSEM (version 1.3.1; *Li and Dewey, 2011*). The DEG analysis was performed using the edgeR R package (version 3.32.1; *Robinson et al., 2010*). Batch information was added as confounding variables to adjust for batch effects. The DEGs are summarized in *Supplementary file 10*.

## Processing of ATAC-seq data

Each ATAC-seq dataset was processed using the ENCODE ATAC-seq pipeline implemented with Caper (*Jin-Wook et al., 2018*). Briefly, reads were mapped to the human reference genome (GRCH38/hg38) using Bowtie2 (version 2.3.4.3), and unmapped reads, duplicates, and those mapped to the mitochondrial genome were removed. Peaks were called by MACS2 (*Zhang et al., 2008*), and optimal peaks that were reproducible across pseudo replicates were used in the downstream analysis. The number of processed reads and peaks is summarized in *Supplementary file 9*. Plots of ATAC-seq signals around the TSSs of expressed genes were generated using the R genomation package (version 1.22.0; *Akalin et al., 2015*). The batch effects of the signals were corrected using the *removeBatchEffect* function from the limma R package (version 3.46.0; *Ritchie et al., 2015*). Of the broad and narrow peaks resulting from the ENCODE ATAC-seq pipeline, the latter were used as an input to obtain

consensus ACRs using the diffBind R package (version 3.0.15; *Ross-Innes et al., 2012*). The dACRs were detected using the edgeR R package (version 3.32.1; *Robinson et al., 2010*). In total, 70,746 ACRs and 5219 dACRs were detected in HeLa cells and are summarized in *Supplementary file 10*. The genomic positions of the ACRs were annotated using the ChIPseeker R package (version 1.26.2; *Yu et al., 2015*). If the ACRs spanned more than one genomic region, their positions were assigned based on the following priority: promoters 5′ untranslated regions (UTRs) 3′UTRs other exons introns downstream intergenic regions. The promoter of a gene was defined as the region 5 kb upstream and 500 bp downstream of the TSS.

### PCA of ATAC- and RNA-seq data

For ATAC-seq, normalized read counts derived from the diffBind R package (version 3.0.15; *Ross-Innes et al., 2012*) were transformed into a $\log_2$ function. Batch effect corrections were done using the limma R package (version 3.46.0; *Ritchie et al., 2015*). For RNA-seq, counts per million mapped reads (CPM) were also processed in the same manner. For PCA, 2000 features with the highest variance across samples were extracted and utilized. Plots of principal components 1 and 2 were generated using the factoextra R package (version 1.0.7).

### Functional signatures of repressed genes upon TXNIP depletion

Genes that exhibited decreased chromatin accessibility at their promoter and decreased RNA expression upon TXNIP knockdown (*Supplementary file 11*) were selected based on the following criteria: (1) $\log_2$ (RNA level in siTXNIP-treated cells/RNA level in siCon-treated cells) (hereafter, siTXNIP/siCon) $\leq -1$; (2) $\log_2$ (siTXNIP/siCon) of ACRs in the promoter region $\leq -1$ (If there are multiple ACRs in the promoter region, the one with the highest ATAC-seq signal was selected) or $\log_2$ mean (siTXNIP/siCon) of all ACRs in the promoter region $\leq -1$. Enrichment analysis of the GO terms in the gene set was performed using g:Profiler (*Raudvere et al., 2019*). Top 10 enriched terms from the biological process and molecular functions categories were plotted (*Figure 5G*).

### Code availability

All source codes and in-house codes used in the study are available at GitHub (copy archived at *Lee, 2023*).

### Acknowledgements

We thank all of the BIGLab and Kwon Lab members for critical reading and comments. This work was supported by the National Research Foundation (NRF) funded by the Ministry of Science and ICT (2021R1A2C3005835, 2022M3E5F1018502, and RS-2023-00207840 to JWN, 2020R1A6A3A13077354 to KTL) and Korea Basic Science Institute (National Research Facilities and Equipment Center) grant funded by the Ministry of Education (grant no. 2023R1A6C101A009). YK and IP were supported by R35GM128752 to YK from the National Institutes of Health.

## Additional information

### Funding

| Funder | Grant reference number | Author |
| --- | --- | --- |
| National Research Foundation of Korea | 2021R1A2C3005835 | Jin-Wu Nam |
| National Research Foundation of Korea | 2022M3E5F1018502 | Jin-Wu Nam |
| National Research Foundation of Korea | RS-2023-00207840 | Jin-Wu Nam |
| National Research Foundation of Korea | 2020R1A6A3A13077354 | Kyung-Tae Lee |

| Funder | Grant reference number | Author |
|---|---|---|
| Korea Basic Science Institute | 2023R1A6C101A009 | Jin-Wu Nam |
| National Institutes of Health | R35GM128752 | Young V Kwon |

The funders had no role in study design, data collection and interpretation, or the decision to submit the work for publication.

## Author contributions

Kyung-Tae Lee, Resources, Data curation, Software, Formal analysis, Investigation, Visualization, Methodology, Writing – original draft, Writing – review and editing, Computational analysis; Inez KA Pranoto, Resources, Investigation, Writing – original draft, Writing – review and editing; Soon-Young Kim, Hee-Joo Choi, Ngoc Bao To, Validation, Investigation, Writing – original draft, Writing – review and editing; Hansong Chae, Resources, Validation, Writing – original draft; Jeong-Yeon Lee, Jung-Eun Kim, Validation, Writing – original draft, Project administration, Writing – review and editing; Young V Kwon, Conceptualization, Resources, Supervision, Writing – original draft, Project administration, Writing – review and editing; Jin-Wu Nam, Conceptualization, Supervision, Writing – original draft, Project administration, Writing – review and editing

## Author ORCIDs

Kyung-Tae Lee http://orcid.org/0000-0001-5354-6910
Inez KA Pranoto https://orcid.org/0000-0002-0709-5642
Soon-Young Kim http://orcid.org/0009-0001-5132-7797
Hee-Joo Choi http://orcid.org/0000-0001-6432-8193
Jeong-Yeon Lee http://orcid.org/0000-0003-1298-7466
Young V Kwon https://orcid.org/0000-0002-8937-3182
Jin-Wu Nam https://orcid.org/0000-0003-0047-3687

## Ethics

All animal experiments were conducted after obtaining approval from Kyungpook National University (Approval Numbers: KNU-2021-0186).

Reviewer #1 (Public Review): https://doi.org/10.7554/eLife.88328.4.sa1
Reviewer #2 (Public Review): https://doi.org/10.7554/eLife.88328.4.sa2
Reviewer #3 (Public Review): https://doi.org/10.7554/eLife.88328.4.sa3
Author Response https://doi.org/10.7554/eLife.88328.4.sa4

# Additional files

## Supplementary files

• Supplementary file 1. List of α-arrestins from human and *Drosophila*. Information about α-arrestin proteins from human (**A**) and *Drosophila* (**B**).

• Supplementary file 2. Evaluation sets of α-arrestins protein–protein interactions (PPIs). Positive and negative PPIs of α-arrestins for human (**A, B**) and *Drosophila* (**C, D**), respectively.

• Supplementary file 3. Summary tables of Significance Analysis of INTeractome express (SAINTexpress) results. Summary tables of SAINTexpress results for human (**A**) and *Drosophila* (**B**).

• Supplementary file 4. Protein domains and short-linear motifs in the α-arrestin interactomes. Summary of protein domains and short-linear motifs annotated in the interactome of each α-arrestin for human (**A, B**) and *Drosophila* (**D, E**). Annotated interactions between the short-linear motifs in α-arrestins and protein domains in the interactomes from the ELM database are also summarized for human (**C**) and *Drosophila* (**F**).

• Supplementary file 5. Enriched Pfam domains in the α-arrestin interactomes. Results of enrichment test of Pfam domains in the interactome of each α-arrestin for human (**A**) and *Drosophila* (**B**).

• Supplementary file 6. Subcellular localizations of α-arrestin interactomes. Summary tables of subcellular localizations of α-arrestins interactomes for human (**A**) and *Drosophila* (**B**).

- Supplementary file 7. Summary of protein complexes and cellular components associated with α-arrestins. Results of the protein complex enrichment analysis tool (COMPLEAT) and enrichment test of cellular component GO terms for human (**A, B**) and *Drosophila* (**C, D**).

- Supplementary file 8. Orthologous relationship of α-arrestin interactomes between human and *Drosophila*. Predictions of orthologs for α-arrestins and their interacting proteins between human and *Drosophila* using DIOPT (**A, B**). Results of enrichment test of GO terms (biological process, molecular functions, and BP, MF, and KEGG pathway) in the identified orthologs are also summarized for human (**C**) and *Drosophila* (**D**).

- Supplementary file 9. Summary of ATAC- and RNA-seq read counts before and after processing. For ATAC-seq, the number of properly paired reads, filtered/deduplicated reads, and identified narrow peaks is summarized. For RNA-seq, the number of filtered and alignable reads is summarized. *Filtered/dedup reads, filtered/deduplicated reads.

- Supplementary file 10. Differential accessibility of accessible chromatin regions (ACRs) and gene expression. Summary of differential accessibility of ACRs (**A**) and gene expression (**B**) between control and TXNIP-depleted condition in HeLa cells.

- Supplementary file 11. Summary of ATAC-seq peaks located in promoters and gene expression level. (**A**) Profiles of ATAC-seq peaks located in promoters of genes. Changes in peak intensities and gene expression levels are also summarized. (**B**) List of genes that exhibited decreased chromatin accessibility at their promoter and decreased RNA expression upon TXNIP knockdown.

- Supplementary file 12. Authentication of cell lines and mouse-derived bone marrow-derived macrophages (BMMs). The STR analysis report for the HeLa cell line and mycoplasma test results for all cell lines utilized in this study are provided as distinct files. Additionally, the genetic and health monitoring report for the C5BL/6 mouse strain from KOATECH is included.

- MDAR checklist

## Data availability

All AP/MS raw spectral count tables from human and *Drosophila* α-arrestins are summarized in *Figure 1—source data 1*. ATAC-seq and RNA-seq data from HeLa cells treated with siCon and siTXNIP can be downloaded from the Korean Nucleotide Archive (KoNA; KAP220517).

The following dataset was generated:

| Author(s) | Year | Dataset title | Dataset URL | Database and Identifier |
|---|---|---|---|---|
| Nam JW | 2022 | Short-read RNA-seq and ATAC-seq in HeLa cells of control and TXNIP depletion (siRNA) | https://www.kobic.re.kr/kona/search_bioproject?bioproject_id=KAP220517 | KoNA, KAP220517 |

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
