## [Editor Report · eLife assessment]

This study provides a **valuable** resource that documents the protein–protein interactions (PPI) network for α-arrestins in both human and *Drosophila* based on affinity purification/mass spectrometry and the SAINTexpress method followed by a series of bioinformatic and functional assessments. Through these, the authors confirmed the roles of known and novel interactions, including proteins involved in RNA splicing and helicase, GTPase-activating proteins, and ATP synthase. This study represents a **convincing** example of how to adopt comparative molecular interactions and how to interpret the functional implications.

---

## [Referee Report · Reviewer #1 (Public Review)]

The study provides a complete comparative interactome analysis of α-arrestin in both humans and *Drosophila*. The authors have presented interactomes of six humans and twelve *Drosophila* α-arrestins using affinity purification/mass spectrometry (AP/MS). The constructed interactomes helped to find α-arrestins binding partners through common protein motifs. The authors have used bioinformatic tools and experimental data in human cells to identify the roles of TXNIP and ARRDC5: TXNIP-HADC2 interaction and ARRDC5-V-type ATPase interaction. The study reveals the PPI network for α-arrestins and examines the functions of α-arrestins in both humans and Drosophila. The authors have carried out the necessary changes that were suggested.

I would like to congratulate the authors and the corresponding authors of this manuscript for bringing together such an elaborate study on α-arrestin and conducting a comparative study in *Drosophila* and humans.

---

## [Referee Report · Reviewer #2 (Public Review)]

In this manuscript, the authors present a novel interactome focused on human and fly alpha-arrestin family proteins and demonstrate its application in understanding the functions of these proteins. Initially, the authors employed AP/MS analysis, a popular method for mapping protein-protein interactions (PPIs) by isolating protein complexes. Through rigorous statistical and manual quality control procedures, they established two robust interactomes, consisting of 6 baits and 307 prey proteins for humans, and 12 baits and 467 prey proteins for flies. To gain insights into the gene function, the authors investigated the interactors of alpha-arrestin proteins through various functional analyses, such as gene set enrichment. Furthermore, by comparing the interactors between humans and flies, the authors described both conserved and species-specific functions of the alpha-arrestin proteins. To validate their findings, the authors performed several experimental validations for TXNIP and ARRDC5 using ATAC-seq, siRNA knockdown, and tissue staining assays. The experimental results strongly support the predicted functions of the alpha-arrestin proteins and underscore their importance.

---

## [Referee Report · Reviewer #3 (Public Review)]

Lee, Kyungtae and colleagues have discovered and mapped out alpha-arrestin interactomes in both human and *Drosophila* through the affinity purification/mass spectrometry and the SAINTexpress method. Their work revealed highly confident interactomes, consisting of 390 protein-protein interactions (PPIs) between six human alpha-arrestins and 307 preproteins, as well as 740 PPIs between twelve *Drosophila* alpha-arrestins and 467 prey proteins.

To define and characterize these identified alpha-arrestin interactomes, the team employed a variety of widely recognized bioinformatics tools. These analyses included protein domain enrichment analysis, PANTHER for protein class enrichment, DAVID for subcellular localization analysis, COMPLEAT for the identification of functional complexes, and DIOPT to identify evolutionary conserved interactomes. Through these assessments, they not only confirmed the roles and associated functions of known alpha-arrestin interactors, such as ubiquitin ligase and protease, but also unearthed unexpected biological functions in the newly discovered interactomes. These included involvement in RNA splicing and helicase, GTPase-activating proteins, and ATP synthase.

The authors carried out further study into the role of human TXNIP in transcription and epigenetic regulation, as well as the role of ARRDC5 in osteoclast differentiation. It is particularly commendable that the authors conducted comprehensive testing of TXNIP's role in HDAC2 in gene expression and provided a compelling model while revising the manuscript. Additionally, the quantification of the immunocytochemistry data presented in Figure 6 convincingly supports the authors' hypothesis.

Overall, this study holds important value, as the newly identified alpha-arrestin interactomes are likely aiding functional studies of this protein group and advance alpha-arrestin research.

---

## [Author Response]

The following is the authors’ response to the previous reviews.

Reviewer #3 comment

1. One suggestion for improvement is to consider incorporating the results from Figure S9 into in the main Figure 6, which would enhance readers' comprehension.

We appreciate your valuable feedback. Based on the reviewer’s suggestion, we have incorporated results from the Figure S9 into the main Figure 6, as shown below. Manuscripts and figure legends have also been modified accordingly.

**Author response image 1. sa4fig1:**